# A glucose-starvation response regulates the diffusion of macromolecules

Ryan P Joyner[1], Jeffrey H Tang[2], Jonne Helenius[3], Elisa Dultz[1†], Christiane Brune[1], Liam J Holt[4], Sebastien Huet[5], Daniel J Müller[3], Karsten Weis[1,2]*

[1]Department of Molecular and Cell Biology, University of California, Berkeley, Berkeley, United States; [2]Institute of Biochemistry, Department of Biology, ETH Zurich, Zürich, Switzerland; [3]Department of Biosystems Science and Engineering, ETH Zurich, Zürich, Switzerland; [4]Institute for Systems Genetics, New York University School of Medicine, New York, United States; [5]CNRS, UMR 6290, Institut Génétique et Développement, University of Rennes, Rennes, France

**Abstract** The organization and biophysical properties of the cytosol implicitly govern molecular interactions within cells. However, little is known about mechanisms by which cells regulate cytosolic properties and intracellular diffusion rates. Here, we demonstrate that the intracellular environment of budding yeast undertakes a startling transition upon glucose starvation in which macromolecular mobility is dramatically restricted, reducing the movement of both chromatin in the nucleus and mRNPs in the cytoplasm. This confinement cannot be explained by an ATP decrease or the physiological drop in intracellular pH. Rather, our results suggest that the regulation of diffusional mobility is induced by a reduction in cell volume and subsequent increase in molecular crowding which severely alters the biophysical properties of the intracellular environment. A similar response can be observed in fission yeast and bacteria. This reveals a novel mechanism by which cells globally alter their properties to establish a unique homeostasis during starvation.

*For correspondence: karsten. weis@bc.biol.ethz.ch

Present address: [†]Institute of Biochemistry, Department of Biology, ETH Zurich, Zürich, Switzerland

## Introduction

Eukaryotic cells expend a significant amount of energy to establish and maintain a high degree of intracellular organization in order to regulate and orchestrate their complex metabolism. This includes the active enrichment of macromolecules into organelles, which in turn allows for the separation of biochemical pathways and enhances their efficiency by increasing local concentrations of enzymes and metabolites. Similarly, active transport by motor proteins on the cytoskeleton enables large eukaryotic cells to overcome the limits of Brownian motion in which the diffusion time scales with the square of the distance. Nevertheless, many cellular pathways and biochemical interactions are dependent on Brownian diffusion. Much research has therefore been performed to characterize diffusional processes that occur within eukaryotic cells. These studies have revealed that the very complex and extremely crowded cytosol of eukaryotic cells cannot be viewed as an ideal liquid but, instead, can be modeled as a polymer gel (*Luby-Phelps, 2000*; *Clegg, 1984*; *Knull and Minton, 1996*) or soft colloidal glass (*Fabry et al., 2001*; *Mandadapu et al., 2008*; *Luby-Phelps, 2000*). Still, a comprehensive understanding of the biophysical properties of eukaryotic cells is lacking, and it remains poorly understood how these characteristics influence macromolecular movement. Furthermore, little is known about the biological determinants of intracellular diffusion including whether cells functionally regulate their diffusional properties in response to changes in growth conditions or the environment.

**eLife digest** Most organisms live in unpredictable environments, which can often lead to nutrient shortages and other conditions that limit their ability to grow. To survive in these harsh conditions, many organisms adopt a dormant state in which their metabolism slows down to conserve vital energy. When the environmental conditions improve, the organisms can return to their normal state and continue to grow.

The interior of cells is known as the cytoplasm. It is very crowded and contains many molecules and compartments that carry out a variety of vital processes. The cytoplasm has long been considered to be fluid-like in nature, but recent evidence suggests that in bacterial cells it can solidify to resemble a glass-like material under certain conditions. When cells experience stress they stop dividing and alter their metabolism. However, it was not clear whether cells also alter their physical properties in response to changes in the environment.

Now, Joyner et al. starve yeast cells of sugar and track the movements of two large molecules called mRNPs and chromatin. Chromatin is found in a cell compartment known as the nucleus, while mRNPs are found in the cytoplasm. The experiments show that during starvation, both molecules are less able to move around in their respective areas of the cell. This appears to be due to water loss from the cells, which causes the cells to become smaller and leads to the interior of the cell becoming more crowded. Joyner et al. also observed a similar response in bacteria. Furthermore, Joyner et al. suggest that the changes in physical properties are critical for cells to survive the stress caused by starvation.

A separate study by Munder et al. found that when cells become dormant the cytoplasm becomes more acidic, which causes many proteins to bind to each other and form large clumps. Together, the findings of the studies suggest that the interior of cells can undergo a transition from a fluid-like to a more solid-like state to protect the cells from damage when energy is in short supply. The next challenge is to understand the molecular mechanisms that cause the physical properties of the cells to change under different conditions.

A particularly well-studied model of intracellular movement is the motion of chromatin in the nucleus. Various groups have analyzed chromatin mobility, but a coherent mechanistic understanding of this process has yet to emerge (reviewed in *Hübner and Spector, 2010*). For example, it was shown that chromosomes wiggle in a manner consistent with constrained diffusion and it was proposed that chromosome movement results from Brownian motion rather than from active motility (*Marshall et al., 1997*). In contrast, other studies have suggested that chromatin movement is ATP dependent (*Heun et al., 2001*), and it was hypothesized that chromatin movement is driven by a multitude of ATP-dependent processes along the length of the chromosome (*Neumann et al., 2012*). Similarly, the role of the cytoskeleton in chromatin movement has remained controversial. For example, both microtubule-dependent and -independent movement was reported (*Heun et al., 2001*, *Marshall et al., 1997*), and several recent studies have also implicated the actin cytoskeleton in chromatin mobility (*Chuang et al., 2006*; *Koszul et al., 2008*; *Spagnol and Dahl, 2014*; *Spichal et al., 2016*).

To better characterize the diffusional properties of eukaryotic cells, we have analyzed here the movement of chromatin and mRNPs in budding yeast under changing growth conditions. Our results demonstrate that cells dramatically change their biophysical properties in response to glucose starvation, which causes a confinement of macromolecules and affects the mechanical properties of the cell. This effect cannot be explained by changes in ATP levels or pH but can be induced by a loss of cell volume without any change in cell mass. This response seems to be conserved as bacteria similarly restrict macromolecular mobility in response to starvation (*Parry et al., 2014*), and we show here that starvation also induces a volume loss in bacterial cells and severely affects the rigidity of fission yeast cells. Our results suggest a novel mechanism by which cells regulate their biophysical properties in order to adapt to environmental stress.

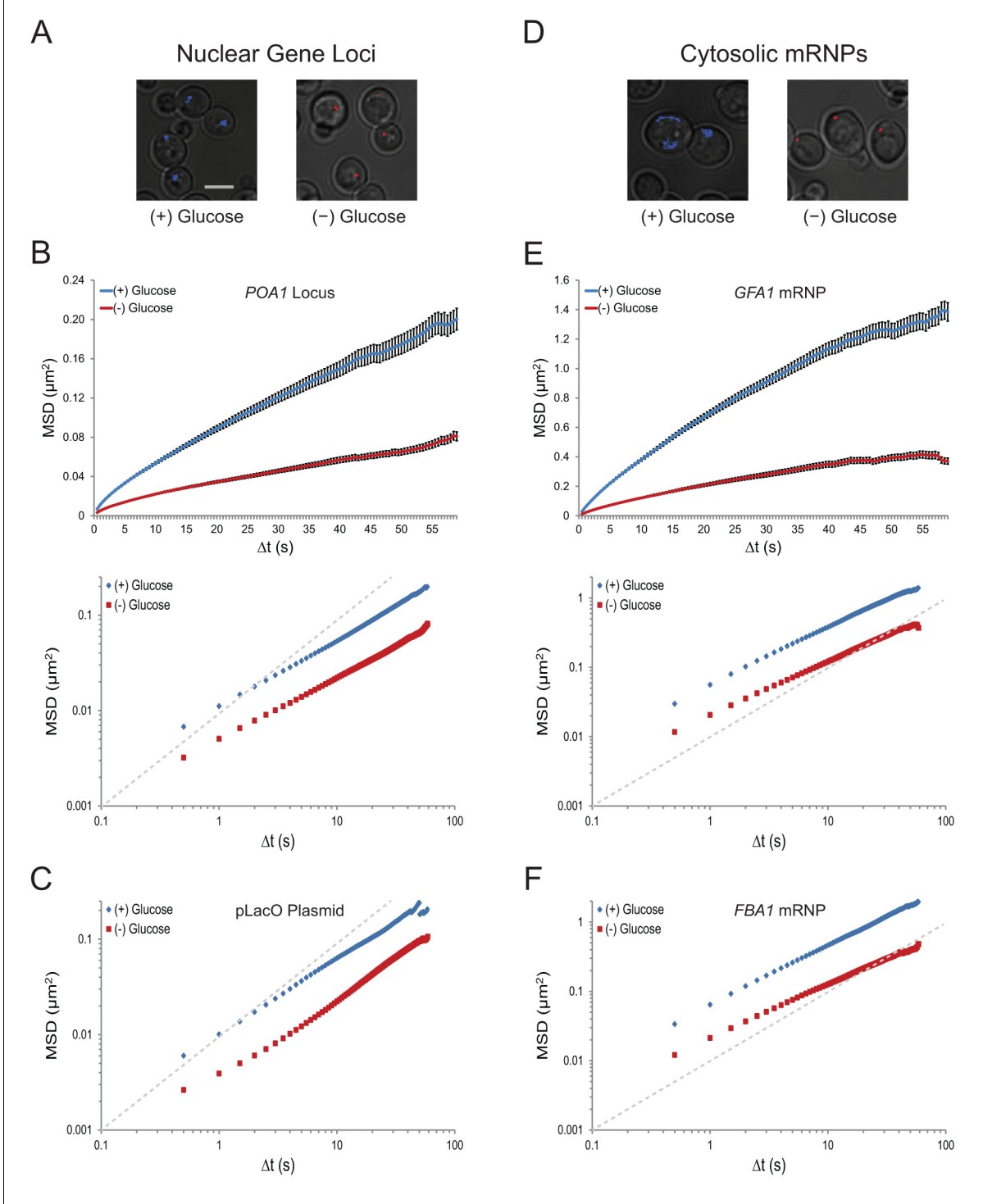

**Figure 1.** Acute glucose starvation confines macromolecular mobility in the nucleus and cytoplasm (*Figure 1—figure supplement 1*). (**A**) Minute-long trajectories of the *POA1* locus from both (+) glucose (blue) and (–) glucose (red) conditions projected on bright field images. Log-growing cells in (+) glucose were acutely starved for glucose, (–) glucose, for 30 min minutes prior to imaging. Scale bar: 4 µm. (**B**) Mean square displacement (MSD) curves for *POA1* mobility. Upper panel: individual MSDs were averaged into an aggregate MSD for each condition. Error bars represent standard error of the mean (SEM). Lower panel: log-log MSD plot of the same data. (**C**) Log-log MSD plot of the pLacO plasmid during exponential growth and after acute glucose starvation. (**D**) Minute-long trajectories of *GFA1* mRNPs from both (+) glucose (blue) and (–) glucose (red) conditions projected on bright field images. (**E**) Mean square displacement (MSD) curves for *GFA1* mRNP mobility. Upper panel: individual MSDs were averaged into an aggregate MSD for

*Figure 1 continued on next page*

*Figure 1 continued*

each condition. Error bars represent SEM. Lower panel: log-log MSD plot of the same data. (**F**) Log-log MSD plot of the *FBA1* mRNP during exponential growth and after acute glucose starvation. Dashed gray lines represent a slope of one to guide the eye.

The following figure supplements are available for figure 1:

**Figure supplement 1.** Glucose starvation affects the mobility of nuclear and cytoplasmic objects.

**Figure supplement 2.** Starvation confines macromolecular mobility.

# Results

## Glucose starvation limits macromolecular mobility in the nucleus and cytoplasm

To examine the effects of changes in growth conditions on nuclear chromatin dynamics in budding yeast, we analyzed the movement of various gene loci. LacO repeats were integrated at the *POA1* locus on chromosome II, the *URA3* locus on chromosome V, and on a centromeric plasmid (pLacO). Co-expression of LacI-GFP allowed us to visualize these three loci and track their mobility over minute-long sequences. Whereas several changes in growth conditions, including growth in different carbon sources or nitrogen starvation, had no obvious effect on chromatin mobility (data not shown), acute glucose starvation induced a dramatic cessation of chromatin movement (*Figure 1A*). This suggests that chromatin mobility is regulated by the presence of glucose.

To quantify the dramatic changes in chromatin mobility, we calculated ensemble-averaged mean square displacements (MSDs) for the chromatin loci (n = 183–1172 trajectories each) (*Figure 1B and C*; *Figure 1—figure supplement 1A*; *Figure 1—figure supplement 2A*). These plots express the magnitude of diffusion for a given particle, quantifying the average displacement per unit time and are used to compute their effective diffusion coefficients (*Qian et al., 1991*). We find that the confinement of chromatin upon glucose starvation (*Figure 1B and C*; *Figure 1—figure supplement 2*)

**Table 1.** Effective diffusion coefficients (K; $\mu m^2$/s) and anomalous diffusion exponents ($\alpha$) for macromolecules in each condition.

| Condition | POA1 Locus | | pLacO Plasmid | | URA3 Locus | | GFA1 mRNP | | FBA1 mRNP | |
|---|---|---|---|---|---|---|---|---|---|---|
| | K | $\alpha$ | K | $\alpha$ | K | $\alpha$ | K | $\alpha$ | K | $\alpha$ |
| (+) Glucose | 0.0057 | 0.69 | 0.0067 | 0.78 | 0.0076 | 0.65 | 0.0420 | 0.83 | 0.0501 | 0.85 |
| (-) Glucose | 0.0023 | 0.64 | 0.0021 | 0.80 | 0.0022 | 0.73 | 0.0131 | 0.77 | 0.0139 | 0.77 |
| (-) Glucose pH 7.4 | 0.0015 | 0.65 | – | – | – | – | 0.0120 | 0.75 | – | – |
| DMSO | 0.0059 | 0.56 | 0.0046 | 0.70 | 0.0060 | 0.61 | 0.0491 | 0.83 | 0.0541 | 0.83 |
| Nocodazole | 0.0040 | 0.48 | 0.0025 | 0.57 | 0.0046 | 0.51 | 0.0364 | 0.85 | 0.0397 | 0.85 |
| Latrunculin A | 0.0038 | 0.50 | 0.0024 | 0.63 | 0.0038 | 0.55 | 0.0476 | 0.82 | 0.0550 | 0.81 |
| Nocodazole + LatA | 0.0028 | 0.48 | 0.0014 | 0.49 | 0.0026 | 0.52 | 0.0303 | 0.81 | 0.0367 | 0.82 |
| 2 mM K$^+$Sorbate | 0.0056 | 0.73 | 0.0051 | 0.80 | 0.0059 | 0.68 | 0.0402 | 0.82 | 0.0296 | 0.80 |
| 4 mM K$^+$Sorbate | 0.0050 | 0.75 | 0.0044 | 0.72 | 0.0048 | 0.71 | 0.0406 | 0.78 | 0.0242 | 0.79 |
| 6 mM K$^+$Sorbate | 0.0039 | 0.70 | 0.0018 | 0.66 | 0.0027 | 0.66 | 0.0378 | 0.76 | 0.0270 | 0.78 |
| 8 mM K$^+$Sorbate | 0.0023 | 0.64 | 0.0012 | 0.61 | 0.0014 | 0.61 | 0.0340 | 0.76 | 0.0164 | 0.77 |
| 0.4 M NaCl | 0.0030 | 0.69 | 0.0026 | 0.75 | – | – | 0.0129 | 0.83 | 0.0146 | 0.85 |
| 0.6 M NaCl | 0.0012 | 0.60 | 0.0011 | 0.58 | – | – | 0.0047 | 0.83 | 0.0057 | 0.84 |
| 0.8 M NaCl | 0.0009 | 0.63 | 0.0011 | 0.63 | – | – | 0.0013 | 0.67 | 0.0016 | 0.77 |
| Quiescence | – | – | – | – | – | – | 0.0004 | 0.29 | 0.0015 | 0.68 |
| 0.02% Azide (Wash) | 0.0037 | 0.74 | – | – | – | – | 0.0293 | 0.82 | – | – |
| 0.02% Azide (Spike) | 0.0012 | 0.67 | – | – | – | – | 0.0155 | 0.81 | – | – |

leads to an approximately three-fold reduction of the apparent diffusion coefficient (K): for instance, $K_{POA1}$ decreased from 5.7 x 10$^{-3}$ μm$^2$/s to 2.3 x 10$^{-3}$ μm$^2$/s upon starvation (*Table 1*). The change in mobility at this time scale was not caused by a change in the anomaly of the diffusion process as the anomalous diffusion exponent (α), which is given by the slope of the curves in the MSD log-log plot, is not affected (see also *Table 1*).

To analyze whether glucose starvation uniquely affects chromatin dynamics in the nucleus, or whether it also influences the mobility of other macromolecules, we imaged the movement of cytoplasmic mRNPs, which can be conveniently tracked as single particles (*Shav-Tal et al., 2004*). 24-PP7 stem-loops were integrated into the 3′ UTR of *GFA1* and *FBA1*, essential genes involved in distinct processes (*Lagorce et al., 2002*; *Schwelberger et al., 1989*), and the movement of individual mRNPs was examined upon co-expression of the coat-binding protein, CP-PP7-3xYFP. Cumulative track projections revealed substantially higher mobility for mRNPs than chromosomal loci in glucose, which is expected given the significantly smaller size of mRNPs compared to chromosome fibers (*Figure 1D*) (*Thompson et al. 2010*; *Zarnack and Feldbrügge, 2007*). Yet, upon glucose starvation, *GFA1* and *FBA1* mRNPs also exhibited a dramatic reduction in their mobility (*Figure 1E and F*; *Figure 1—figure supplement 1B*). Removal of glucose led to a three- to four-fold decrease in the diffusion coefficient of both *GFA1* ($K_{GFA1}$ reduced from 4.2 x 10$^{-2}$ μm$^2$/s to 1.3 x 10$^{-2}$ μm$^2$/s) and *FBA1* ($K_{FBA1}$ reduced from 5.0 x 10$^{-2}$ μm$^2$/s to 1.4 x 10$^{-2}$ μm$^2$/s) mRNPs without affecting the anomalous exponent (α). These values are similar to the relative change that we observed in the diffusion of chromatin (*Table 1*). Depletion of glucose by growth into quiescence also confined the mobility of mRNPs; therefore, the effects we observe are not a consequence of our cell washes (*Figure 1—figure supplement 2B and C*). The decrease in mobility of both chromosomal loci and mRNP particles suggests that glucose starvation causes a confinement of macromolecules in the nucleus as well as the cytoplasm.

## Starvation arrests cytoskeletal dynamics, which constrains chromatin mobility but has little effect on mRNPs

To begin to understand the nature and mechanism of the starvation-induced reduction in chromatin and mRNP mobility, we first focused on the cytoskeleton. In eukaryotes, the movement of many macromolecules is directly influenced by the dynamics of the cytoskeleton. For instance, mRNPs can be actively transported by motor proteins along the cytoskeleton and both the actin cytoskeleton and microtubules were reported to influence chromatin dynamics (*Thompson et al., 2010*; *Heun et al., 2001*; *Koszul et al., 2008*; *Spagnol and Dahl, 2014*; *Spichal et al., 2016*). Interestingly, actin filaments were also shown to rapidly depolymerize upon starvation (*Uesono et al., 2004*; *Sagot et al., 2006*). We therefore wanted to explore whether the starvation-induced confinement of macromolecular mobility resulted from changes in cytoskeletal dynamics. Indeed, our starvation condition lead to ablation of actin filaments in nearly all cells (*Figure 2A*) and a nearly four-fold reduction of microtubule elongation events in G1 cells as visualized by Tub1-GFP (*Figure 2B*).

If the changes in the mobility of chromatin or mRNPs upon glucose withdrawal were due to the reduction of cytoskeletal dynamics, we would predict that drug-induced inhibition of actin and/or microtubule polymerization mimics the starvation response. In order to test this hypothesis, we treated cells with the actin depolymerizer, latrunculin A (LatA), and/or the microtubule depolymerizing drug, nocodazole (*Ayscough et al., 1997*; *Jacobs et al., 1988*). Indeed, treatment with either drug reduced chromatin movement, although the effect was insufficient to mimic the confinement of chromatin during starvation (*Figure 2C*; *Figure 2—figure supplement 1*). Concurrent treatments with LatA and nocodazole reduced chromatin mobility in an additive manner to substantially reduce chromatin mobility (*Figure 2C*; *Figure 2—figure supplement 1*). Moreover, log-log MSD plots indicate that the effect of inhibiting the cytoskeleton is most apparent at longer time intervals and may augment the anomalous behavior of chromatin motion ($α_{POA1\text{-}DMSO}$ = 0.56 versus $α_{POA1\text{-}Noc+LatA}$ = 0.48; *Figure 2C*; *Figure 2—figure supplement 1*; *Table 1*).

The actin and microtubule cytoskeleton could either act on chromatin itself or affect chromatin mobility indirectly by influencing overall nuclear motion. To differentiate between these two possibilities, we tracked the distance between two labelled chromosomal loci (*POA1*, chromosome II and *PES4*, chromosome VI) over time (*Figure 2—figure supplement 1*). This approach allows for the exclusive quantification of intranuclear movements because translational changes in the positioning of the nucleus affect both loci identically and thus, do not influence intranuclear distance

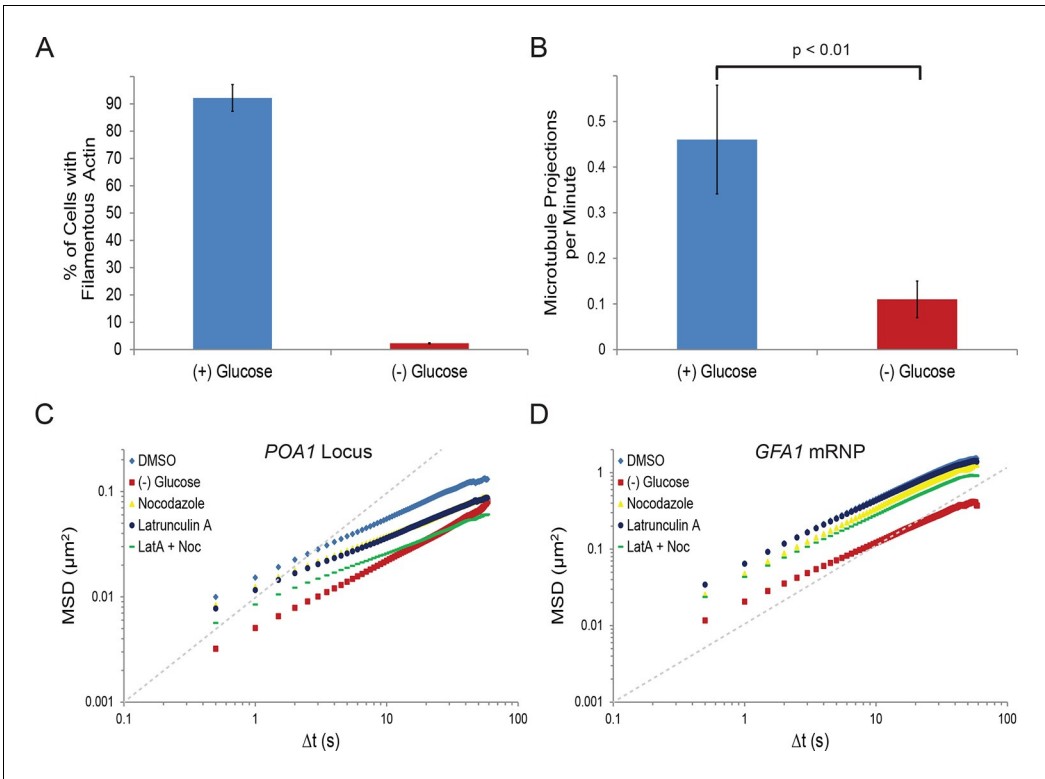

**Figure 2.** Starvation confines both cytoskeleton-independent macromolecular mobility and mobility influenced by the cytoskeleton (*Figure 2—figure supplement 1*). (**A**) Quantification of filamentous actin during logarithmic growth, (+) glucose, and after acute starvation, (–) glucose. Cells were fixed and stained with phalloidin. Z-stack projections were then processed and cells were classified based on the presence or absence of filamentous actin. Error bars are standard error for three biological replicates (n = 406–709 cells per replicate). (**B**) Average number of microtubule projections per G1 cell during logarithmic growth, (+) glucose, and after acute starvation, (–) glucose. Error bars are standard deviation (SD) of the mean from three biological replicates (n = 10–1510-15 cells per replicate). The p-value for a two-tailed t-test for unpaired values assuming equal variance is shown. (**C**) Log-log MSD plot of the *POA1* locus after treatment with nocodazole and/or latrunculin-A (LatA) for 20 min prior to imaging. (**D**) Log-log MSD plot of the *GFA1* mRNP after treatment as described in (**C**). Dashed gray lines represent a slope of one to guide the eye.

The following figure supplement is available for figure 2:

**Figure supplement 1.** Starvation confines the cytoskeleton-independent mobility of mRNPs and the cytoskeleton-influenced mobility of chromatin.

measurements (*Marshall et al., 1997*). This two-locus analysis confirmed the LatA and nocodazole-induced reduction in chromatin mobility as the interchromosomal distance between the two loci still decreased considerably with simultaneous drug treatment (*Figure 2—figure supplement 1*). Of note, however, this result cannot differentiate between cytoplasmic or nuclear cytoskeletal dynamics in modulating chromatin mobility (e.g. via deformations of the nucleus). In conclusion, actin and microtubule dynamics independently contribute to the mobility of yeast chromatin.

In contrast to chromatin mobility, mRNP mobility was only moderately affected by perturbation of cytoskeletal dynamics (*Figure 2D*; *Figure 2—figure supplement 1*) suggesting that the mobility of the *GFA1* and *FBA1* mRNPs is largely independent of the cytoskeleton. Overall, our results show that glucose starvation restricts cytoskeleton-independent mobility as well as the mobility of macro-molecules influenced by the cytoskeleton.

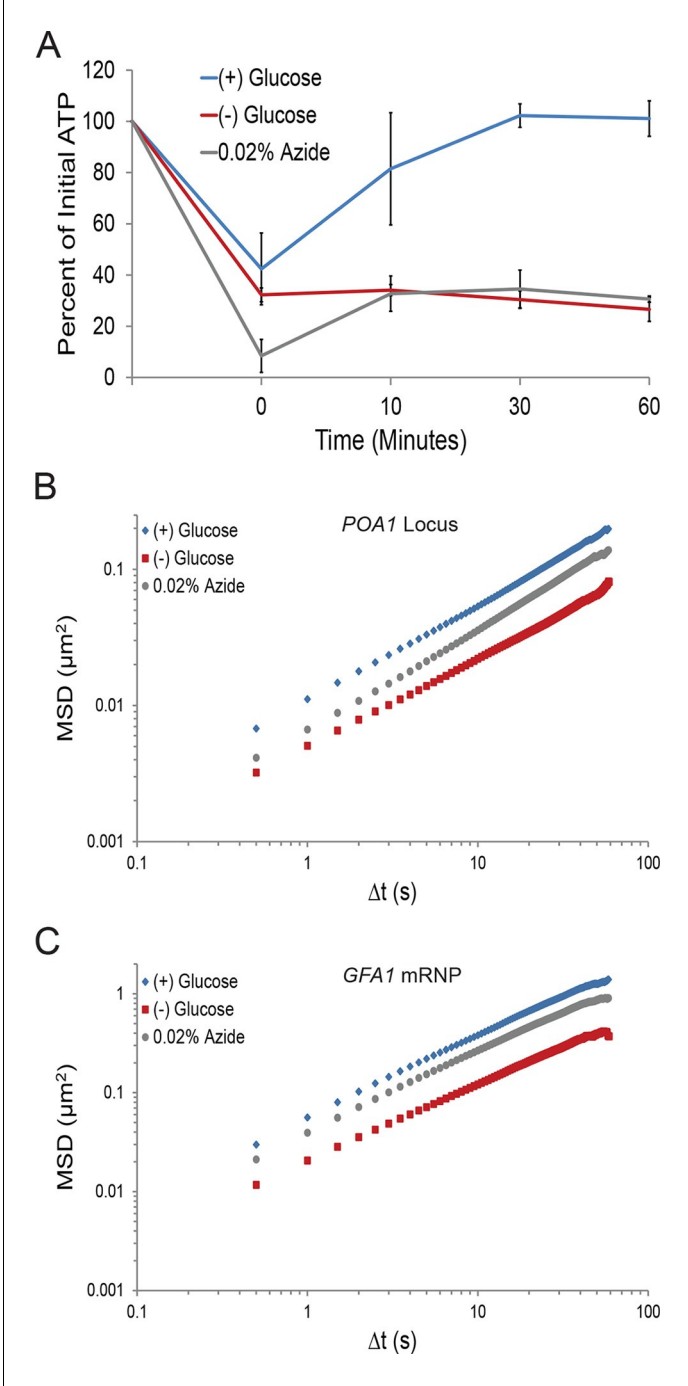

**Figure 3.** A ~70% reduction of intracellular ATP is insufficient to replicate the macromolecular confinement of glucose starvation (*Figure 3—figure supplement 1*). (**A**) Intracellular ATP concentrations of acutely glucose-starved yeast were back-diluted into media containing 2% dextrose (n = 2 experiments), 2% dextrose + 0.02% azide (n = 2 experiments), or maintained in (–) glucose media (n = 3 experiments). Intracellular ATP concentrations were determined using a luciferase-based ATP assay (*Ashe et al., 2000*) and normalized to pre-treatment levels. The zero min time point was taken immediately after back-dilution into the described media. Error bars represent SD. (**B**) Log-log MSD plot of the *POA1* locus. Cells were treated with azide as described in (**A**). The azide treatment fails to replicate the confinement of glucose-starved cells from *Figure 1B*. (**C**) Log-log MSD plot of the *GFA1* mRNP. Cells were treated with azide as described in (**A**). The azide treatment fails to replicate the confinement of glucose-starved cells from *Figure 1E*.

*Figure 3 continued on next page*

*Figure 3 continued*

The following figure supplement is available for figure 3:

**Figure supplement 1.** Respiration maintains intracellular ATP concentrations after acute glucose starvation.

## Reduction of ATP is insufficient to explain the macromolecular confinement

Our results so far could be explained by two alternative models: 1) starvation impacts macromolecular diffusion through multiple, distinct mechanisms, or 2) a singular, starvation-induced pathway restricts the mobility of macromolecules, and leads to both the collapse of cytoskeletal dynamics and the restriction of mRNP mobility. The acute withdrawal of glucose in fermenting yeast cells is expected to have dramatic consequences on cellular physiology. For example, the cellular ATP concentration drops (*Ashe et al., 2000*) and the intracellular pH decreases in starved cells (*Orij et al., 2009*). We therefore tested whether these global changes in cellular physiology lead to the observed changes in macromolecular mobility.

First, we investigated the changes in intracellular ATP concentration during starvation. Upon glucose starvation, the ATP concentration rapidly decreased by ~70%. Remarkably, after this initial drop, ATP levels were relatively stable at ~30% of the initial concentration for the remainder of the experiment (*Figure 3A*). Of note, the maintenance of this reduced ATP level required oxidative phosphorylation as the cellular ATP concentration quickly dropped to nearly undetectable levels when cells deficient in mitochondrial function were starved (*cbp2Δ*; *Shaw and Lewin, 1997*) (*Figure 3—figure supplement 1*). In contrast, when cells washed in sugar-free medium were back-diluted into media containing glucose, ATP levels recovered to the initial concentration within 30. Thus, glucose starvation induces a rapid ~70% reduction of intracellular ATP levels in agreement with previously published results (*Özalp et al., 2010*).

To determine whether a ~70% reduction in intracellular ATP levels is sufficient to confine cytoskeleton-independent macromolecular mobility as well as mobility influenced by the cytoskeleton, we reproduced this decrease in ATP concentrations in the presence of glucose. This was achieved by back-diluting washed cells into glucose media containing 0.02% sodium azide ($NaN_3$), an inhibitor of oxidative phosphorylation (*Figure 3A*). After 0.02% azide treatment, ATP levels dropped to the same level as in glucose-starved cells, but the mobility of chromatin and mRNPs did not decrease to the same degree as in starvation (*Figure 3B and C*). We therefore conclude that a ~70% reduction of intracellular ATP, as observed in glucose-depleted cells, is insufficient to fully explain the confinement of macromolecular mobility during glucose starvation.

## Reduction of intracellular pH titrates macromolecular mobility but cannot explain starvation-induced macromolecular confinement

Next, we investigated the contribution of intracellular pH to macromolecular confinement upon starvation. The intracellular pH ($pH_i$) of budding yeast was reported to decrease from pH ~7.3 to pH ~6.4 in a time frame consistent with the observed confinement of macromolecular mobility upon glucose deprivation (*Orij et al., 2009*; *Young et al., 2010*). Using the pH-sensitive GFP analog, pHluorin, as a $pH_i$ biosensor (*Miesenböck et al., 1998*), we confirmed in our experimental conditions that glucose starvation causes a reduction in $pH_i$ from ~7.4 to ~6.4 (*Figure 4A*). We then manipulated intracellular proton concentrations by the addition of the weak acid potassium sorbate ($K^+$Sorbate). $K^+$Sorbate traverses the plasma membrane, releases a proton, and reduces $pH_i$ in a concentration-dependent manner (*Bracey et al., 1998*; *Piper et al., 2001*). Varying the extracellular concentration of $K^+$Sorbate from 0 mM to 8 mM enabled us to titrate $pH_i$ from pH 7.4 to pH 5.9, thus covering more than the $pH_i$ range observed in cells grown either in (+) glucose or (–) glucose conditions, with little effect on intracellular ATP levels (*Figure 4A and B*). Remarkably, exposing cells to increasing concentrations of $K^+$Sorbate induced macromolecular confinement, and both chromatin and mRNP mobility could be titrated with decreasing $pH_i$ (*Figure 4C–F*; *Figure 4—figure supplement 1*).

Importantly, however, treatment with the highest concentration of $K^+$Sorbate (8 mM) was necessary to fully recapitulate the starvation-induced confinement of both chromatin and mRNPs

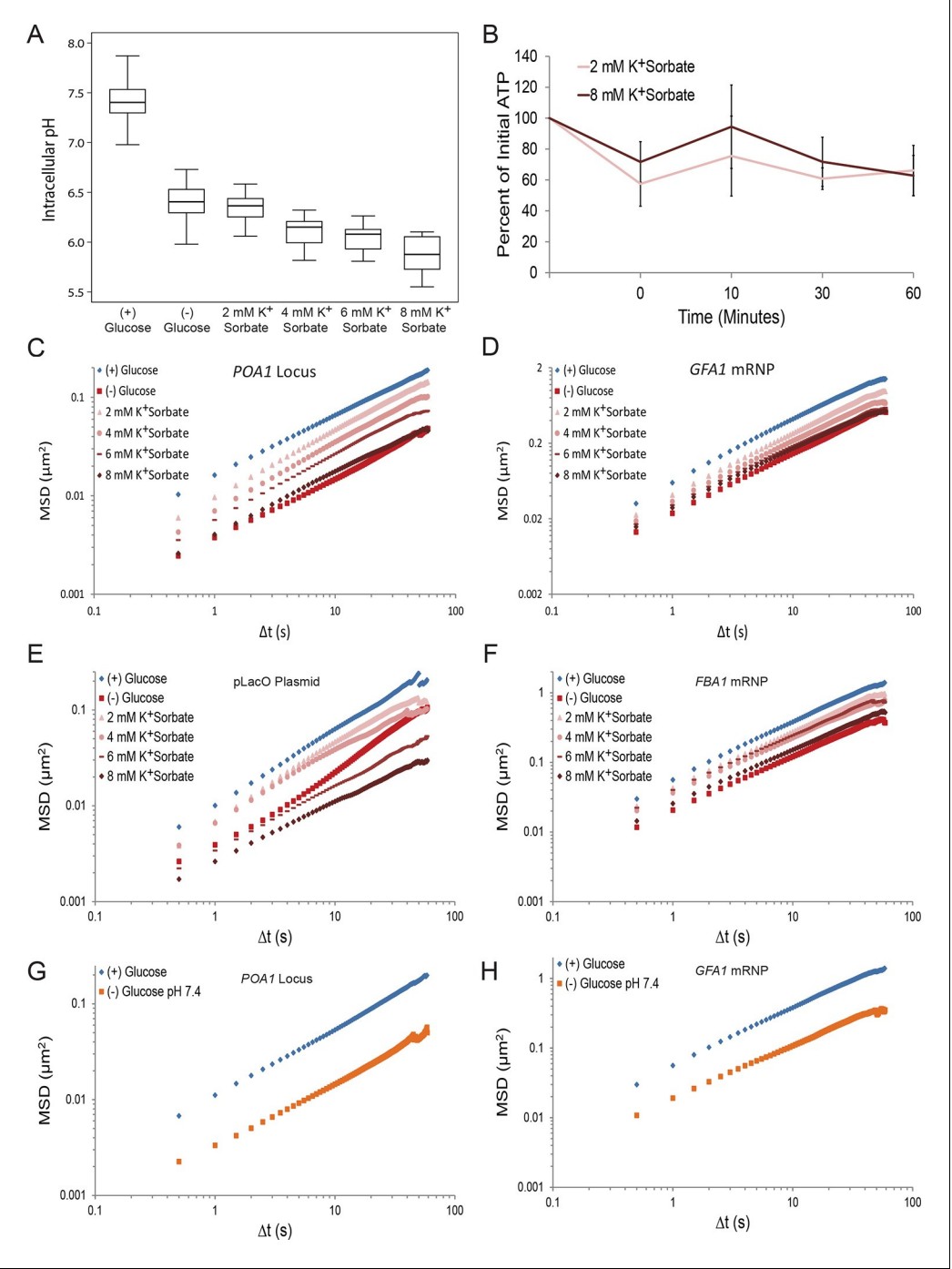

**Figure 4.** A drop in intracellular pH (pH$_i$) can reduce macromolecular mobility but cannot explain the confinement observed in glucose-starved cells (**Figure 4—figure supplements 1** and **2**). (**A**) Boxplots of the pH$_i$ of cells acutely starved for glucose or treated with varying concentrations of potassium sorbate (K$^+$Sorbate). Log-growing yeast cells expressing phluorin (**Miesenböck et al., 1998**) were acutely starved of glucose or treated with K$^+$Sorbate for 30 min before ratiometric imaging. Intracellular pH was then estimated from a calibration curve (see 'Materials and methods'; **Figure 4—figure supplement 1**). Values from three biological replicates were pooled and compiled into boxplots. Whiskers represent the minimum and maximum respectively. (**B**) Intracellular ATP concentrations after treatment with either 2 mM or 8 mM K$^+$Sorbate. Intracellular ATP concentrations were determined as in **Figure 3A**. The zero minute time point was taken immediately after each treatment. Error bars represent SD (n =2). (**C**) Log-log MSD plot of the *POA1* locus after treatment with K$^+$Sorbate as described in (**A**). (**D**) Log-log MSD plot of the *GFA1* mRNP after treatment as described in (**A**). **E**) Log-log MSD plot of the pLacO plasmid after

*Figure 4 continued on next page*

*Figure 4 continued*

treatment as described in (**A**). (**F**) Log-log MSD plot of the *FBA1* mRNP after treatment as described in (**A**). (**G**) Log-log MSD plot of the *POA1* locus after acute glucose starvation into starvation media (SC) adjusted to pH 7.4. (**H**) Log-log MSD plot of the *GFA1* mRNP after treatment as described in (**G**).

The following figure supplements are available for figure 4:

**Figure supplement 1.** A drop in intracellular pH (pH$_i$) titrates macromolecular mobility.

**Figure supplement 2.** Sodium azide induces a pleiotropic reduction in intracellular pH, which may explain the subsequent confinement of macromolecular mobility.

(*Figure 4C–F*). Under this condition, pH$_i$ dropped to 5.9, which is below the physiologically observed pH$_i$ of 6.4 in glucose-starved cells (*Figure 4A*). When pH$_i$ was lowered to 6.4 (by the addition of 2 mM K$^+$Sorbate), the movement of both chromatin and mRNPs was only moderately reduced (*Figure 4C–F*). Furthermore, starvation in (–) glucose media adjusted to pH 7.4, which prevents a drop in intracellular pH (*Dechant et al., 2010*), failed to inhibit the confinement of chromatin and mRNPs upon glucose starvation (*Figure 4G and H*). Thus, the starvation-induced reduction of pH$_i$ is neither sufficient nor necessary and cannot fully explain the observed confinement of macromolecular mobility.

## Starvation induces a reduction in cellular volume

In the course of our experiments, we found that the nuclear volume decreased upon glucose withdrawal (*Figure 5A*). Since nuclear volume and cellular volume are generally tightly linked (*Neumann and Nurse, 2007*; *Jorgenesen et al., 2002*), we utilized a Coulter Counter to examine whether cell size changes upon glucose starvation. Indeed, the median cell volume decreased from 101.6 fL ($\sigma$ = 54.6) in glucose to 86.1 fL ($\sigma$ = 45.6) in starved cells, corresponding to a volume reduction of ~15% (*Figure 5B*). In addition, we observed that the yeast vacuole, an organelle involved in various processes including protein degradation and metabolite storage, swelled in size under glucose starvation conditions (*Figure 5C*). In non-starved cells, the vacuole-to-cell volume ratio was 0.25 ± 0.02, whereas for starved cells this ratio increased to 0.40 ± 0.01 (mean ± standard error for three independent experiments) (*Figure 5D*). In combination, this vacuolar volume expansion together with the decrease in total cell and nuclear volume, reduces the cytoplasmic space that is available for diffusing molecules upon glucose starvation by ~30% ('Materials and methods').

## A reduction in cellular volume is sufficient to confine macromolecular mobility

We hypothesized that such a large reduction of the accessible cellular volume may be sufficient to induce the observed macromolecular confinement, for example, by an increase in macromolecular crowding or changes in cellular viscosity (*Luby-Phelps, 2000*). If the hypothesis was correct that a decrease of cell volume was the singular response that confines macromolecular mobility, two predictions could be made: (1) lowering the pH$_i$ by the addition of K$^+$Sorbate might induce a decrease in cellular volume since high concentrations of potassium sorbate can replicate the confinement of both chromatin and mRNPs during starvation, and (2) manipulation of cellular volumes, for example, by changes in osmolarity should recapitulate the effects of glucose starvation on cytoskeleton-independent mobility (mRNPs) and on mobility influenced by the cytoskeleton (chromatin).

To test our first prediction, we explored the effect of pH$_i$ changes on cell volume. Remarkably, lowering pH$_i$ by the addition of K$^+$Sorbate also induced an increasing reduction in cell volume. Strikingly, addition of 8 mM K$^+$Sorbate, which led to the same mobility decrease as glucose starvation, reduced the median cell volume to 87.1 fL ($\sigma$ = 46.3), a level virtually identical to that of glucose-starved cells (*Figure 5—figure supplement 1A* and *Figure 5B*). Although intracellular acidification led to a decrease in cell volume, it was not required for this decrease since starvation in (–) glucose media adjusted to pH 7.4, preventing an intracellular pH drop (*Dechant et al., 2010*), also reduced the cell volume (*Figure 5—figure supplement 1B*). Thus, the effects of K$^+$Sorbate are better

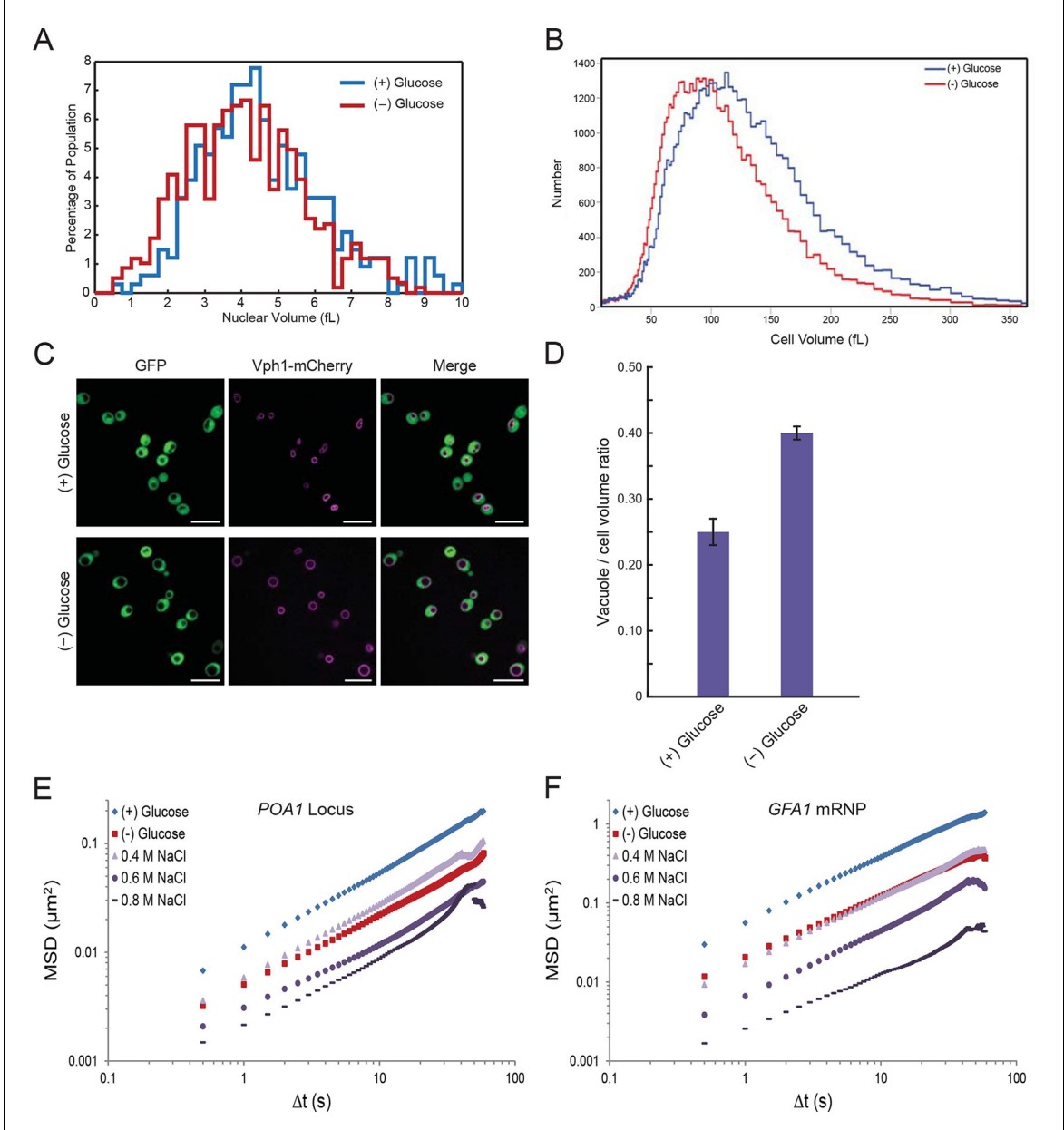

**Figure 5.** Starvation induces a constriction in cell size and an expansion in vacuolar volume (*Figure 5—figure supplements 1* and *2*). (A) Nuclear volume after acute glucose starvation. Histograms of nuclear volumes measured by reconstruction from three-dimensional image stacks using Imaris. The p-value resulting from a two-tailed t-test on the average volume in each condition (unpaired values assuming equal variance) is p<0.001. (B) Histograms of cell volumes of log-growing and acutely starved yeast cells. Log-growing cells, (+) glucose, were acutely starved of glucose, (–) glucose, and cell volume was measured using a Beckman Coulter Multisizer 3 (see 'Materials and methods'). Approximately 50,000 cells were measured for each condition. (C) Cytoplasmic free GFP (green) and vacuolar membrane protein Vph1-mCherry (magenta) fluorescence images. Scale bar: 10 µm. (D) Quantification of vacuole-to-cell volume ratio. Error bars represent standard errors about the mean (N = 3 independent experiments with > 55 cells per experiment). (E) Log-log MSD plot of the *POA1* locus after treatment with increasing concentrations of NaCl. Cells were imaged approximately 10 min after hyperosmotic shock. (F) Log-log MSD plot of the *GFA1* mRNP after treatment as described in (E).

The following figure supplements are available for figure 5:

**Figure supplement 1.** Histograms of cell volumes after various treatments.

**Figure supplement 2.** Hyperosmotic shock increasingly confines macromolecular mobility without affecting intracellular ATP concentrations or intracellular pH.

explained in terms of volume decrease than pH changes. These data suggest a strong relationship between cell size and macromolecular mobility (*Figure 4C–F*; *Figure 5—figure supplement 1*).

To further test our model, we manipulated the cellular volume by changing the extracellular osmolarity. We treated cells with increasing concentrations of NaCl and monitored the effects on chromatin and mRNPs. Notably, addition of NaCl had little effect on pH$_i$ or cellular ATP levels, allowing us to uncouple the effects of cell volume from these factors (*Figure 5—figure supplement 2*). As shown in *Figure 5E and F* and *Figure 5—figure supplement 2C and D*, both chromatin and mRNP mobility were progressively confined by increasing the magnitude of osmotic shock, consistent with recent results demonstrating that osmotic shock affects intracellular diffusion (*Babazadeh et al., 2013*; *Miermont et al., 2013*). Therefore, a reduction of cell volume is sufficient to explain the starvation-induced confinement of macromolecular mobility.

## The reduction in volume leads to an increase in molecular crowding

We next explored how decreased cell volume could impede the movement of macromolecules. In the very densely packed environment of the cell, a reduction in the volume accessible to diffusing molecules can alter protein-protein interactions and induce complex phase changes such as liquid-to-gel-like or liquid-to-glass-like transitions (*Doliwa and Heurer, 1998*; *Weeks et al., 2000*). To test whether glucose starvation causes an increase in molecular crowding, we examined whether the observed volume loss is compensated by a reduction in the total biomass of the cell. We performed lyophilization mass measurement experiments in which starved and non-starved yeast cells were freeze-dried to remove their water content, and their masses were measured and normalized against the total number of cells (*Figure 6A*). We saw no significant difference between the non-starved and starved samples (11.5 ± 0.5 pg/cell and 11.9 ± 0.4 pg/cell, respectively; mean ± standard deviation for three independent experiments). Importantly, yeast cells grown continuously in raffinose, which are known to be smaller than cells grown on glucose (*Tyson et al., 1979*) and thus served as a positive control, showed a 23% reduction in cell mass (8.9 pg/cell) compared to cells grown in glucose. Thus, acute glucose starvation does not induce a decrease in cell mass despite a significant reduction in the accessible cellular volume. This increase in mass density suggests that the intracellular environment experiences increased molecular crowding conditions, which may lead to broad changes in the material properties of the cell.

## Glucose starvation affects the mechanical properties of budding yeast cells

To examine whether the starvation-induced changes in macromolecular mobility are accompanied by changes in the viscoelastic properties of cells we measured the effective stiffness of budding yeast upon acute glucose starvation using atomic force microscopy (AFM). Our setup used wedged AFM cantilevers for parallel plate compression to characterize yeast cell morphology. These live-cell micro-compression experiments provide measurements of cellular viscoelasticity and cell stiffness under various conditions (*Stewart et al., 2013*; *Ramanathan et al., 2015*). After removal of the cell wall, glucose-starved and non-starved yeast spheroplasts were subjected to AFM measurements. Force-response curves were recorded while compressing single cells and analysis showed that glucose starvation induced a change in the cells' mechanical properties increasing their mean stiffness from 18 to 22 nN·µm$^{-1}$ as well as increasing the cell-to-cell variability in stiffness (*Figure 6B*). Consistent with the cell size measurements described above (*Figure 5B*), these AFM experiments also suggest that starved yeast cells are smaller than the corresponding control cells and that there appears to be a negative correlation between cell size and stiffness (*Figure 6—figure supplement 1*).

In a separate approach, we also investigated the rigidity of the cytoplasm by examining cell morphology after removal of the cell wall. Without an intact cell wall, the cytoplasmic turgor pressure causes the yeast spheroplast to 'ball up' into a sphere. Since budding yeast cells are naturally round to begin with, we treated cells with the mating pheromone α-factor to generate shmoos, mating projections that cause the cell to become substantially elongated. Spheroplasted shmooed cells which were not starved of glucose behaved as expected and formed spherical spheroplasts (*Figure 6C*, *Video 1*). Intriguingly, starved shmooed spheroplasts remained predominantly in an elongated form (~four-fold less spherical spheroplasts in starved versus non-starved conditions) (*Figure 6C*, *Video 2*). This resistance to the internal turgor pressure suggests that the starved yeast

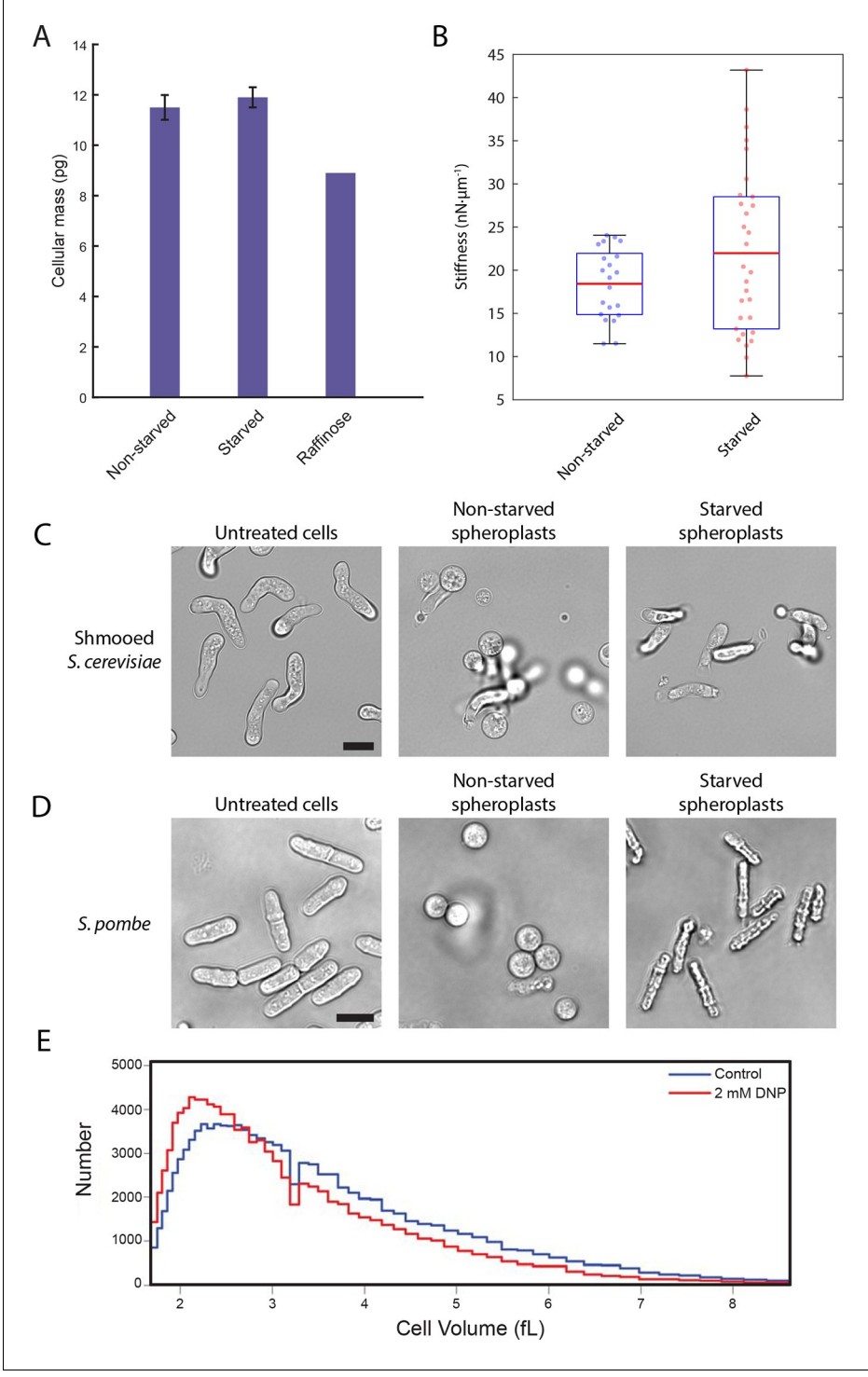

**Figure 6.** A conserved glucose starvation response alters the mechanical properties of the cytoplasm (*Figure 6—figure supplements 1* and *2*). (**A**) Dry mass measurements of non-starved and starved yeast. Yeast cells in normal glucose, glucose deprivation, and raffinose growth conditions were lyophilized, and their cellular dry masses were determined. Error bars represent standard deviations about the mean (N = 3 independent experiments for non-starved and starved conditions; N = 1 for raffinose growth condition). (**B**) AFM experiments measure a mean stiffness increase from 18 to 22 nN·μm⁻¹ for non-starved and starved cells, respectively, along with an increase in cell-to-cell variability (Kolmogorov-Smirnov test, p = 0.013). Box plot: red line represents the mean, blue lines represent the 1ˢᵗ and 3ʳᵈ quartiles, and whiskers represent the minimum and maximum values. Each data point

*Figure 6 continued on next page*

*Figure 6 continued*

represents the mean value of eight measurement cycles on a single cell (see 'Materials and methods'). (C) Mating pheromone-treated budding yeast cells (shmoos) remain elongated after starvation. Shmooed budding yeast cells prior to spheroplasting (left), non-starved shmooed spheroplasts (center), and starved shmooed spheroplasts (right). Scale bar: 10 µm. (D) Fission yeast cells remain elongated after starvation. Fission yeast cells prior to spheroplasting (left), non-starved spheroplasts (center), and starved spheroplasts (right). Scale bar: 10 µm. Spheroplasting efficiency in the experiments described in (B), (C), and (D) was equally efficient in starved and non-starved cells and assessed via lysis in water. (E) Histograms of cell volumes of log-growing wildtype *E. coli* after 30 min of treatment with 2 mM DNP or the solvent control (EtOH). Approximately 100,000 cells were measured in each condition using a Beckman Coulter Multisizer 3.

The following figure supplements are available for figure 6:

**Figure supplement 1.** Cell stiffness negatively correlates with cell size.

**Figure supplement 2.** The actin cytoskeleton is a major consumer of intracellular ATP.

cytoplasm is significantly more rigid than that of non-starved yeast, consistent with our AFM measurements. Furthermore, this stiffening is reversible as re-addition of glucose to the starved spheroplasts returned them to a spherical form (*Video 3*), suggesting that sensory feedback pathways may exist to actively regulate this stiffening mechanism. Taken together our results demonstrate that budding yeast cells undergo dramatic biophysical changes upon acute glucose starvation affecting their viscoelastic properties in addition to their intracellular macromolecular diffusion.

## Changes in biophysical properties are a conserved starvation response

To examine whether similar biophysical changes can also be observed in other unicellular eukaryotes upon glucose withdrawal, we monitored the starvation-response of the fission yeast, *Schizosaccharomyces pombe*. Budding and fission yeast separated evolutionarily several hundred million years ago, and they are as different from each other as either is from metazoans (*Sipiczki, 2000*). Since *S. pombe* is naturally rod-shaped, we performed analogous spheroplasting experiments as described in *Figure 6C*. Upon digestion of the cell wall, *S. pombe* cells round up and become spherical (*Figure 6D*, *Video 4*). Conversely, glucose-starved cells maintain their rod-shape, reminiscent of the shmooed budding yeast cells, again demonstrating that glucose-withdrawal leads to a remarkable stiffening of the cytoplasm (*Figure 6D*, *Video 5*). As for budding yeast, the stiffening is also reversible as addition of glucose to starved *S. pombe* spheroplasts returns them to a spherical shape (*Video 6*).

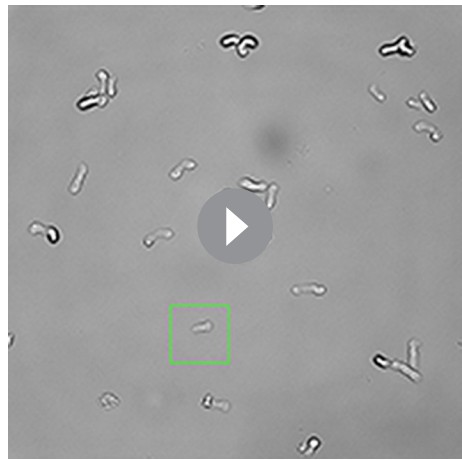

**Video 1.** Spheroplasted budding yeast shmoos.

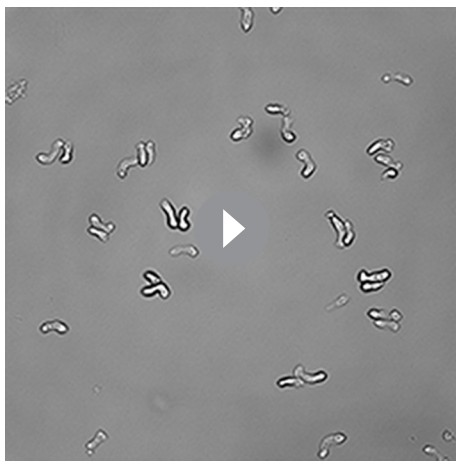

**Video 2.** Spheroplasted starved budding yeast shmoos.

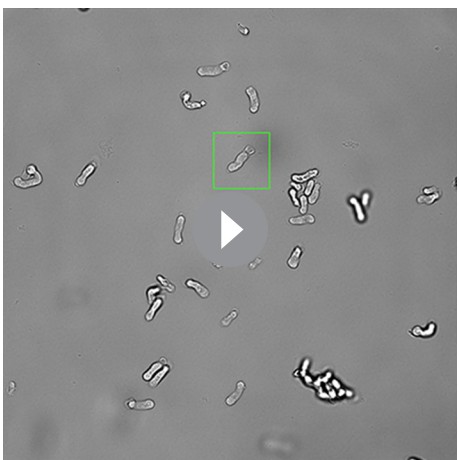

**Video 3.** Re-addition of glucose to spheroplasted
starved budding yeast shmoos.

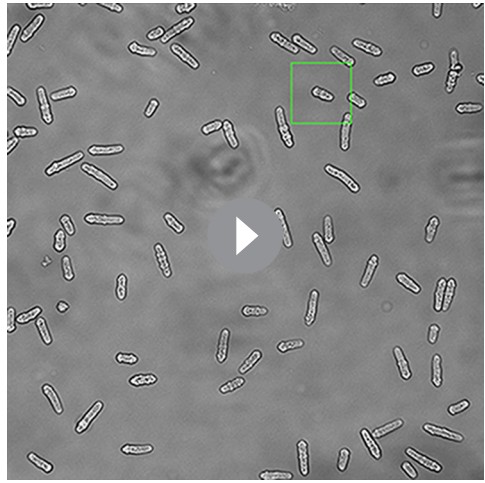

**Video 4.** Spheroplasted fission yeast.

It was recently reported that the bacterial cytoplasm has glass-like properties and can undergo liquid-like to solid-like transitions upon starvation or treatment with 2,4-Dinitrophenol (DNP), an uncoupler of oxidative phosphorylation (*Parry et al., 2014*). This condition results in confinement of macromolecules and disproportionally affects large particles above ~30 nm (*Parry et al., 2014*). The underlying mechanism for this cytoplasmic transition into a solid-like state in bacteria remains unexplored, but *Parry et al. (2014)* suggested that metabolic activity is needed to keep the bacterial cytoplasm fluid. Although the cytoplasmic organization of prokaryotic and eukaryotic cells have dramatic differences, the similarity between these observations in bacteria and our findings in yeast prompted us to test whether there is a common underlying mechanism and whether bacteria also use a reduction in cell volume to increase crowding and confine macromolecular mobility. Strikingly, a 30 min treatment of *E.coli* with 2 mM DNP caused a significant reduction in the volume of *E.coli*, inducing a ~9.5% modal volume contraction (SD = ± 4.1%) (*Figure 6E*). It should be noted, however, that the treatment utilized here, for technical reasons, is longer than the DNP-incubation time reported by *Parry et al. (2014)*. Nevertheless, this result suggests that the reported liquid-to-solid like phase transition of the bacterial cytoplasm may also be triggered by cellular volume loss and suggests that the starvation-induced reduction of cell size could be a deeply conserved starvation response.

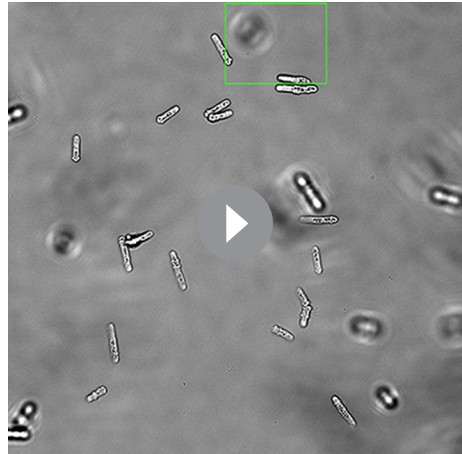

**Video 5.** Spheroplasted starved fission yeast.

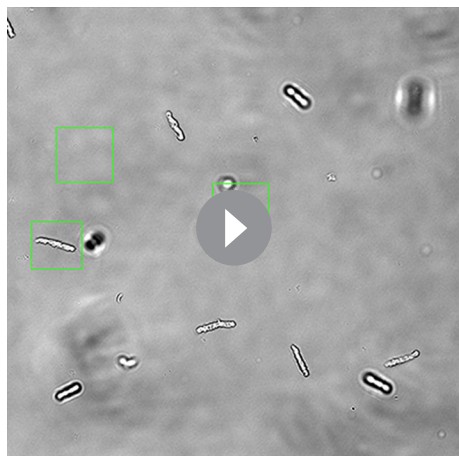

**Video 6.** Re-addition of glucose to spheroplasted
starved fission yeast.

## Discussion

Our results identify a novel cellular response that severely alters the biophysical properties of cells and restricts intracellular diffusion in response to glucose starvation. The dramatic macromolecular confinement observed upon starvation of budding yeast cannot be explained by a reduction of intracellular ATP (*Figure 3*), by a drop in pH (*Figure 4*) or inhibition of global metabolism, as respiration persists upon glucose depletion (*Figure 3—figure supplement 1*). Instead, we propose that increased macromolecular crowding induced by an acute volume loss fundamentally alters the mechanically properties of cells leading to macromolecular confinement. Our results further suggest that both eukaryotic and prokaryotic cells regulate the level of crowding in order to control intracellular diffusion and macromolecular interactions. In this context, it is intriguing that metazoan cells also seem to regulate their cellular volume as it was recently reported that tissue culture cells rapidly swell during cell division, which might aid with the mitotic separation of large chromosomes (*Zlotek-Zlotkiewicz et al., 2015*; *Son et al., 2015*).

At this point, it is unknown how glucose starvation induces a reduction in cellular volume and more work is needed to elucidate the signaling pathways and mechanisms by which glucose starvation causes cell size contraction and vacuolar volume increase. Upon acute glucose starvation, budding yeast cells undergo a notable cytosolic acidification lowering their intracellular pH to ~6.4, presumably by the rapid inhibition of the major plasma membrane proton pump Pma1 (*Figure 4*; *Orij et al., 2009*; *Young et al., 2010*; *Dechant et al., 2010*). Whereas this acidification might contribute to macromolecular confinement and enhance changes of biophysical properties within cells (*Fels et al., 2009*), pH drop alone is neither necessary nor sufficient to recapitulate the macromolecular confinement, and the volume loss observed in glucose starved cells can occur in the absence of acidification (*Figure 4* and *Figure 5—figure supplement 1*). An increase of the intracellular proton concentration below 6 by the addition of increasing concentrations of $K^+$Sorbate is able to induce a complete confinement of macromolecules (*Figure 4C–F*) but also significantly reduces the available cytoplasmic volume. Interestingly, inhibition of cellular metabolism by 2-deoxyglucose and antimycin also provokes an enhanced drop of intracellular pHto 5.5, which in turn triggers macromolecular confinement and leads to profound changes in the mechanical properties of the cell consistent with a sol to gel-like transition of the cytoplasm (Munder *et al.*, personal communication).

Our data also support a mechanism of chromatin mobility in which the movement of yeast chromatin is strongly influenced by the joint dynamics of both actin and microtubules. Precedence for such a mechanism is evidenced by previous work describing similar microtubule-dependent motion and reported interactions between microtubules, spindle pole bodies, and centromeres throughout the cell cycle (*Heun et al., 2001*; *Tanaka and Tanaka, 2009*). Moreover, the actin cytoskeleton is known to propel the movement of meiotic chromosomes and to influence the diffusion of other large macromolecules, suggesting a central function throughout the life cycle of yeast (*Koszul et al., 2008*; *Brangwynne et al., 2009a*; *Zhou et al., 2011*). The long-range directional movement of chromosomal loci is also reported to be dependent on the actin cytoskeleton (*Chuang et al., 2006*). This is consistent with our data, as the cytoskeletal inhibitors latrunculin-A and nocodozale appeared to more strongly affect the mobility of chromatin at longer timescales and have minor effects at shorter ones (*Figure 2*; *Figure 2—figure supplement 1*). Finally, recent work detailing the existence of actin filaments in the nucleus substantiates the possibility of a network of nuclear actin (*Baarlink et al., 2013*). However, because latrunculin A depolymerizes all actin, our work cannot currently differentiate between the effects of cytoplasmic and nuclear actin in modulating chromatin mobility.

Paradoxically, previous experiments exploring the ATP dependence of chromatin mobility reached opposing conclusions in that treatment with sodium azide had only minor effects on the mobility of chromosomes in one study (*Marshall et al., 1997*) but drastically reduced their mobility in another (*Heun et al., 2001*). By tracking both chromatin and mRNPs in conditions that closely replicated the perceived differences in the two published protocols (*Marshall et al., 1997*; *Heun et al., 2001*), we were indeed able to duplicate the distinct results (*Figure 4—figure supplement 2*). Importantly, however, we also observed that azide treatment lowered $pH_i$ and that the distinct treatments reduced intracellular pH to dramatically different extents (*Figure 4—figure supplement 2*). Thus, a pleiotropic effect of azide treatment on $pH_i$ might be sufficient to reconcile the observed differences in the confinement of macromolecules. These results also show that the azide treatment used here (*Figure 3* (wash); *Figure 4—figure supplement 2*) replicates both the decrease in

intracellular ATP concentrations and the drop in pH$_i$ in glucose-starved cells. This indicates that changes in ATP and pH$_i$ do not have additive effects and are neither alone nor in combination sufficient to explain the effects on macromolecular mobility that we observe in glucose starvation.

As expected (*Luby-Phelps, 2000*), our single particle-tracking experiments show that both chromatin and mRNPs display anomalous subdiffusion at the timescales that we investigated. Interestingly, while glucose starvation dramatically decreases the diffusivity of the tracked objects, it does not change the anomaly of the diffusion. Anomalous subdiffusion is usually considered to be a transient phenomenon that occurs in complex, disordered media such as the intracellular environment (*Höfling and Franosch, 2013*). At short timescales, objects explore a pure viscous environment thus displaying normal Brownian motion. Yet, at intermediate timescales, the motion of the objects is hindered by static structures, leading to anomalous subdiffusion. Finally, at long timescales, that is, when the traveled distances are much larger than the sizes of the structures that impede movement, diffusion is again pure Brownian motion. The fact that glucose deprivation affects diffusivity but not anomaly suggests that starvation induces a dramatic increase of the 'microscopic' viscosity without affecting the higher order cellular structures that impede macromolecule motion (*Huet et al., 2014*). Based on our results and according to the classical molecular crowding theory (*Zimmerman and Minton, 1993*) this change of microscopic viscosity is probably the consequence of an increase in crowding upon cell size contraction.

As the cell cytosol becomes more viscous and restrictive to diffusional motion under glucose starvation, we posit that it must also become more resistive to external perturbations on the cytoplasm. Indeed, we find that upon starvation the cytoplasm of budding and fission yeast undertake an astonishing transition leading to an increase in the viscoelastic stiffness of the cell (*Figure 6*). Our observations are reminiscent of recently published observations in bacteria describing a glassy liquid-to-solid transition upon glucose starvation or DNP treatment (*Parry et al., 2014*). Furthermore, we demonstrate that bacteria also undergo a volume loss similar to starved yeast (*Figure 6E*). This observation suggests that molecular crowding also triggers the reported phase transition in bacteria, and similar mechanisms might be activated in both pro- and eukaryotes in response to glucose starvation.

Notably, this modulation of intracellular diffusion represents a mechanism to globally regulate the biochemistry of cells. As described for bacteria (*Parry et al., 2014*), we envision glucose starvation as a dormant state which might hold cells in stasis until provisions in the environment improve. Consequently, we propose that cells functionally regulate intracellular diffusion in order to establish a cellular organization that is restrictive to some molecular interactions, perhaps thereby conserving energy, while simultaneously permissive to essential biochemistry.

The conservation of cell contraction between prokaryotes and eukaryotes strongly suggests a vital function during starvation. On the one hand, the reduction in macromolecular mobility may allow for the rapid inhibition of excessive energy expenditure by diminishing unnecessary molecular interactions. This is consistent with previous observations describing the rapid inhibition of principal ATP consumers upon starvation. For example, Pma1p, required for regulation of intracellular pH and thought to use upwards of 20% of cellular ATP, is immediately inhibited upon glucose starvation (*Bracey et al., 1998*). Ribosomes also disengage from mRNA, resulting in a global collapse of polysome profiles (*Ashe et al., 2000*). Additionally, we found that depolymerization of the actin cytoskeleton may function, in part, to conserve ATP, as treatment with latrunculin A rapidly increases intracellular ATP concentrations (*Figure 6—figure supplement 2*). Because osmotic shock promptly inhibits translation initiation and depolymerizes the actin cytoskeleton, it is conceivable that the starvation-induced cytoskeletal breakdown and collapse of polysome profiles, as well as other starvation-associated phenotypes, could be partly driven by molecular crowding (*Chowdhury et al., 1992*; *Ueseono et al., 2004*; *Ashe et al., 2000*; *Uesono and Toh-EA, 2002*). Therefore, the confinement of intracellular diffusion through molecular crowding may prove vital for economical biochemistry and represent a very upstream event upon carbon starvation.

On the other hand, molecular crowding-enhanced aggregation and confined macromolecular mobility may promote sustained interactions between complexes during starvation. In agreement with this hypothesis, starvation is known to induce the formation of compartmentalized biochemistry and potential pockets of liquid-liquid unmixing (*Narayanaswamy et al., 2009*; *Brangwynne et al., 2009b*). For example, P-bodies, macromolecular complexes composed of mRNPs and mRNA decay machinery, form in the absence of glucose and potentially aggregate through unstructured regions

of the component proteins (*Ramachandran et al., 2011*). Moreover, stationary phase induces the formation of foci and filaments for large numbers of proteins (*Narayanaswamy et al., 2009*). This aggregation was hypothesized to enhance interactions between members of the same pathway and promote more efficient biochemistry (*Narayanaswamy et al., 2009*) but was more recently also shown to cause enzymatic inactivation in some instances (*Petrovska et al., 2014*). As a consequence, prolonged starvation and confinement of macromolecular mobility may also pose unique challenges to cells, as complexes with elongated stretches of unstructured regions may prove more prone to extensive aggregation. It will now be interesting to explore whether other cell types, including metazoan cells, respond similarly to starvation or growth factor depletion and if limited diffusion indeed stimulates aberrant protein aggregation, a process which has been implicated in a wide variety of human diseases (*Ross and Poirier, 2004*).

## Materials and methods

### Strains and growth

Yeast strains used in this study are of the W303 strain-background, except for strains containing labeled mRNPs which are heterozygous diploids of W303 and S288c backgrounds. The *E. coli* strain utilized for volume measurements is wild type BW25113. Genotypes for every strain are listed in *Table 2*; plasmids are described in *Table 3*. Unless otherwise indicated, yeast cells were grown at room temperature in synthetic complete (SC) media at pH 5.0 (titrated with 1 M HCl) containing 2% dextrose (SCD). Specific dropout media was used to maintain plasmids when necessary. *E. coli* were grown at 30°C in LB.

### Imaging and particle tracking

Time-lapse epifluorescence imaging was performed using an inverted Nikon TE2000 microscope equipped with an Andor CCD camera, a motorized stage (Applied Scientific Instrumentation, Inc., Eugene, OR), 60x/1.49NA APO TIRF oil immersion objective, and controlled with Metamorph® software (Molecular Devices, LLC, Sunnyvale, CA). Cells were stabilized by coating the glass-bottom of MatTek® dishes (PG35G-1.5-14-C; MatTek Corporation, Ashland, MA) with 0.2% concanavalin A and aspirating the solution prior to adding 300 µL of $OD_{600}$ 0.2–0.3 cells. All imaging was performed at room temperature. Images for particle tracking were collected at a frame rate of 0.5 s for 60 s, unless otherwise indicated. Particles were tracked using custom MATLAB® (The MathWorks, Inc.,

**Table 2.** Yeast strains used in this study.

| Strain | Genotype | Source |
|---|---|---|
| KWY165 | W303; *MATa ura3-1 leu2-3 his3-11,15 trp1-1 ade2-1* | This Study |
| KWY1622 | W303; *MATα ybr022w::256LacO::LEU2 his3::LacI-GFP::HIS3 trp1::dsRED-HDEL::TRP1* | *Green et al., 2012* |
| KWY3541 | *KWY 1622, cbp2Δ::KANMX* | This Study |
| KWY3538 | W303; *MATα his3::LacI-GFP::HIS3 trp1::dsRED-HDEL::TRP1 256LacO::URA3* | This Study |
| KWY2848 | W303; *MATα his3::LacI-GFP::HIS3 trp1::dsRED-HDEL::TRP1 yel021w::128LacO::URA3* | This Study |
| KWY4586 | W303; *MATa ybr022w::112TetO::URA3 yfr023w::256LacO::LEU2 his3::LacI-GFP_TetR-3XmCherry::HIS3* | This Study |
| KWY970 | *KWY 165, ura3::pHIS-GFP-TUB1::URA3* | This Study |
| KWY3661 | W303; *MATa pHluorin::URA3* | This Study |
| KWY4796 | W303; *MATα trp1::dsRED-HDEL::TRP1 leu2::TetR-GFP::LEU2* | This Study |
| KWY4736 | W303 *MATa/S288c MATα NDC1/ndc1::NDC1-tdTomato::KANMX GFA1/gfa1::GFA1-24PP7 3xYFP-PP7-CFP::HIS3* | This Study |
| KWY4737 | W303 *MATa/S288c MATα NDC1/ndc1::NDC1-tdTomato::KANMX FBA1/fba1::FBA1-24PP7 3xYFP-PP7-CFP::HIS3* | This Study |
| KWY5112 | W303; *MATa ura3-1 leu2-3 his3-11,15 trp1-1 ade2-1 VPH1::VPH1-mCherry::KANMX leu2::GFP::CaURA3* | This Study |
| KWY6241 | CAF13 (*S. pombe* wildtype) | This Study |
| yYB5978 | S288c; *MATa his3Δ1 leu2Δ0 ura3Δ0 met15Δ0 LYS2 ADE2 TRP1 bar1::kanMX* | *Caudron and Barral, 2013* |

**Table 3.** Plasmids used in this study.

| Plasmid | Description | Source |
|---------|-------------|--------|
| pKW2734 | pHMX-256LacO (CEN, URA3) | This Study |
| pKW2957 | pYES2-PACT1-pHluorin (CEN, URA3) | *Orij et al., 2009* |
| pKW544 | MET25$_{pro}$-PP7-CP-3xYFP (CEN, HIS3) | This Study |

Natick, MA) scripts based on the 2D feature finding and particle tracking software made available by Dr. Maria Kilfoil (*Pelletier et al., 2009*; http://people.umass.edu/kilfoil/downloads.html).

## Quantification of macromolecular mobility

An average mean square displacement (MSD) was calculated for each condition using the well-described equation:

$$MSD\tau = \left( r_{(t+\tau)} - r_t \right)^2$$

Effective diffusion coefficients, K, were computed by finding the slope of the best-fit line between the first and tenth time points of the averaged MSD. The anomalous diffusion exponents of the log-log MSDs, α, were determined by fitting the first 40 time intervals of the averaged MSD to a power law.

## Acute starvation

Five milliliters of cells grown in SCD media (OD$_{600}$ = 0.4–0.7) were collected by centrifugation (3000 rpm) for 2 min. The supernatant was then removed and cells resuspended in 1 mL of SC media. Four additional wash steps followed, with two 2 min spins (6000 rpm) succeeded by two 1 min spins. The OD$_{600}$ of the final cell suspension was determined and cells diluted into fresh SC or SCD media for analysis. Time-lapse epifluorescence imaging was performed 30 min after the wash.

## Potassium sorbate treatments

Logarithmic cells growing in SCD media (pH 5.0) were incubated with an appropriate concentration of a stock 20 mM K$^+$Sorbate in SCD (pH 5.0) to give rise to the corresponding final concentration of K$^+$Sorbate. Cells were treated for 30 min before imaging.

## Sodium azide treatments

All experiments involving sodium azide (NaN$_3$) took place in SC or SCD media at pH ~5.8. Cells were treated with a final concentration of 0.02% NaN$_3$ (in water) by two distinct protocols. In one treatment, sodium azide was directly added, or 'spiked', into a logarithmic culture. In the other, cells were first washed (via the acute starvation method) in SC media before being resuspended into SCD media containing 0.02% NaN$_3$. This treatment is referred to as 'wash' and is the condition described in the main text.

## Cytoskeleton perturbations

Cells were treated with either 15 µg/mL nocodazole (from a 1.5 mg/mL stock in DMSO; Sigma-Aldrich, St. Louis, MO), and/or 200 µM latrunculin A (from a 10 mM stock in DMSO; Sigma-Aldrich) for 20 min before imaging. This results in 3% DMSO as the control.

## Osmotic shock

Cells grown in SCD media (OD$_{600}$ = 0.4–0.7) were collected by centrifugation (3000 rpm) for 2 min and resuspended in SCD media containing the appropriate concentration of NaCl (0.4, 0.6, or 0.8 M). Cells were then imaged after 10 min.

## ATP measurements

The protocol employed was only marginally modified from *Ashe et al. (2000)*. Briefly, 20 μL of sample (KWY1622 cells in culture) was added to 20 μL of 10% TCA and immediately vortexed for 1 min. Of the 40 μL solution, 10 μL was then added to 990 μL of ATP reaction buffer (25 mM HEPES, 2 mM EDTA, pH 7.75). After the time course, 50 μL of a luciferin/luciferase kit (ATP Determination Kit, Sensitive Assay; B-Bridge International, Inc., Cupertino, CA) was directly added to 50 μL of the buffered-ATP solutions and briefly mixed by pipetting the solutions in a Corning 96-well opaque white plate (Corning, Inc., Tewksbury, MA). Reactions took place in the dark for 10 min at room temperature. Luminescence was then measured using a Tecan Infinite 200 Pro microplatereader (Tecan Group Ltd., Männedorf, Switzerland) which was set for a 10 s integration time and no attenuation. Sample aliquots were taken at 0, 10, 30 and 60 min and standardized to a pre-treatment control. ATP concentrations were also corrected for $OD_{600}$ at the beginning and end of each time course. ATP standards were used to generate a standard curve to insure linearity and estimate intracellular ATP concentrations.

## CellASIC ONIX imaging of quiescent cells

We utilized a CellASIC ONIX microfluidic profusion platform and accompanying microfluidic plates to track particles in quiescent cells (EMD Millipore, a division of Merck KGaA, Darmstadt, Germany). Quiescence was established by allowing cells to grow within the same SCD media for 7 days (*Laporte et al., 2011*). Seven-day-old media, collected from the culture by centrifuging 1mL of cells and harvesting the supernatant, was used as perfusion media in the microfluidic plate at a rate of 2 psi.

## Cell volume measurements

Cell volume measurements were performed as described in *Goranov et al. (2009)*. In short, 1 mL of cells (KWY4736) were briefly sonicated (1 s, 2X) and diluted (1:100) into Isoton II Dilutent (Beckman Coulter, Inc., Brea, CA) such that saturation levels fluctuated between 1–3% during measurements on a Beckman Coulter Multisizer 3. If oversaturation triggered warnings at any point, the run was terminated and sample repeated. The aperture was also cleaned and flushed between each run. For yeast, the particle count was set to 50,000 and for bacteria, 100,000.

## Filamentous actin quantification

To quantify filamentous actin, KWY165 cells were fixed with formaldehyde at a final concentration of 4% for 15 min. Fixed cells were washed twice with potassium phosphate buffer and resuspended into PBS-BSA. Cells were then stained with Alexa Fluor 568-phalloidin (Invitrogen, Thermo Fisher Scientific, Waltham, MA) and Hoechst stain. Samples were incubated in the dark at room temperature for 1 hr and washed again with PBS-BSA prior to imaging (*Laporte et al., 2011*). Background was subtracted from z-stacks (0.15 μm steps over 3 μm) taken with a 100x/1.49NA APO TIRF oil immersion objective using a rolling ball radius of 1 pixel in ImageJ (NIH, Bathesda, MD). Maximal intensity projections were then blinded and cells classified based on the presence or absence of apparent filamentous or bundled actin.

## Intracellular pH measurements

The phluorin calibration protocol was slightly adapted from *Orij et al., (2009)* to establish a standard curve. 50 μg/mL (in PBS) of digitonin was used to permeabilize cells for 10 min. After washing cells in PBS and placing them on ice, cells were then incubated in citric acid/$Na_2HPO_4$ buffer with pH values ranging from pH 5.0 to pH 8.0. phluorin emission intensity ratios were then calculated from calibration images (taken with a 100x/1.49NA APO TIRF oil immersion objective) using a custom script in ImageJ (NIH) and compared to the pH values of the corresponding buffer. The resulting curve was used to estimate experimental intracellular pH values generated from emission ratios produced from the same ImageJ script. The standard curve and all experiments were performed using KWY3661.

## Nuclear volume measurements

Nuclear volume measurements were conducted on KWY4796 cells expressing TetR-GFP (and DsRed-HDEL). Cells were grown in complete synthetic medium with 2% dextrose overnight at 25°C to OD 0.5–0.8. Cells were then treated as indicated and transferred to Nunc Lab-Tek chambers (Thermo Fisher Scientific) coated with concanavalin A at OD 0.2. To wash cells into different medium conditions, cells were pelleted at 6000 rpm in a table top centrifuge and washed four times with synthetic medium lacking dextrose and then resuspended in synthetic medium either lacking dextrose or supplemented with 2% dextrose. Three-dimensional stacks with 0.24 μm z stepping were acquired on a Zeiss Axio Observer Z1 spinning disk confocal microscope equipped with a motorized piezo stage (Applied Scientific Instrumentation, Inc.) and an EMCCD camera using a 100x 1.46 Oil, alpha Plan Apochromat objective. TetR-GFP was excited with a 488 nm laser line and scanned using a Quad Band 'RQFT' 405/488/568/647 dichroic and a 520/35 band pass emission filter. Image stacks were processed in Imaris (Bitplane AG, Zurich, Switzerland) to reconstruct nuclear volume.

## Microtubule dynamics

*S. cerevisiae* cells with GFP-Tub1 integrated at the *URA3* locus were grown overnight to mid-log phase in SCD media (pH 5.0) and then treated by acute glucose starvation (as above). The cells were then adhered to Concanavalin A treated 384-well glass plates while maintaining the experimental condition. Cells were imaged by epifluorescence time-lapse microscopy for 20 min: a z-stack of five images, each 1 μm apart was acquired once every minute. The resulting images were processed by a maximum intensity projection and the number of microtubule elongation events per minute was manually scored by an experimental blinded to the sample identity.

## Lyophilization mass measurements

KWY165 yeast cells were grown in 400 mL synthetic complete + 2% dextrose (SCD) media and harvested during log-phase growth (OD$_{600}$ ~ 0.8). Cells were spun down, supernatant removed, and cells re-suspended in 50 mL synthetic complete without dextrose (SC) media, and then repeated with two further wash steps. The sample was then finally re-suspended into 4 mL SC media. 2 mL were added to 48 mL SCD media, and 2 mL were added to 48 mL SC media. Both samples were incubated at 25°C for 30 min. Then the samples were spun down, supernatant removed, and cells re-suspended into 1 mL water, which was repeated two more times. After the final wash step, the supernatant was removed, and the cell pellet was lyophilized under 0.040 mbar pressure at -50°C for 24 hr using the Alpha 2–4 LDplus freeze-dryer (Martin Christ GmbH, Osterode am Harz, Germany). The dry mass of the cells were measured, and the cells were re-suspended in water and counted using a hemocytometer to determine the average mass per cell for both the non-starved and starved cells. For mass measurements with raffinose, the protocol was the same except dextrose was replaced with raffinose.

## Vacuolar volume measurements

KWY5112 cells expressing GFP and N-terminally tagged Vph1-mCherry were grown in SCD media and harvested during log-phase growth. The standard acute glucose starvation protocol was performed, and the cells were placed on concanavalin A-coated Lab-Tek chambered slides (Thermo Fisher Scientific Inc, Waltham, MA) and imaged on a spinning disk confocal microscope (Carl Zeiss AG, Jena, Germany). The entire cell volume was imaged using a z-stack (0.24 μm z-slice height) under 488 nm (GFP) and 561 nm (mCherry) laser excitation with a 100x α Plan-Apochromat objective. The z-stack images were segmented and the vacuolar and cellular volumes were computed using custom Python scripts.

## Accessible cytoplasmic volume calculation

To calculate the change in accessible cytoplasmic volume after glucose starvation, the total cell shrinkage (15%), nuclear volume reduction (4.60 fL to 4.12 fL), and vacuolar volume expansion (25 to 40% of cellular volume) were taken into account. The accessible cytoplasmic volumes were computed using:

$$
\begin{aligned}
A_{NS} &= C_{NS} - V_{NS} - N_{NS} \\
A_S &= C_S - V_S - N_S \\
C_S &= 0.85 \cdot C_{NS} \\
V_{NS} &= 0.25 \cdot C_{NS} \\
V_S &= 0.40 \cdot C_S
\end{aligned}
$$

where $A$, $C$, $V$ and $N$ denote the accessible cytoplasmic volume, total cell volume, vacuolar volume, and nuclear volume, respectively, the subscripts $NS$ and $S$ denote the non-starved and starved conditions, and $N_{NS}$ = 4.60 fL and $N_S$ = 4.12 fL.

## AFM cell stiffness assay

For stiffness measurements of yeast cells an AFM (Cellhesion 200, JPK Instruments) was mounted on an inverted microscope (Observer.Z1, Zeiss). DIC images were recorded using a 40x/1.2 C-Apochromat water immersion objective (Zeiss). Tip-less cantilevers (NSC12/noAl, Mikromasch, 350 μm long, nominal spring constant 150 mN·m$^{-1}$) were modified with PDMS (Sylgard 184, Dow Corning) wedges to correct for tilt angle of the cantilever and to be able to confine yeast cells between two parallel plates [*Stewart et al., 2013*]. KWY165 cells in log-phase growth were spheroplasted for 1 hr in 200 μg·mL$^{-1}$ zymolyase 100T (Amsbio) in SCD plus 1 M sorbitol at 30℃. Spheroplasts were then either washed into SCD plus 1 M sorbitol (non-starved treatment) or SC plus 1 M sorbitol (starved treatment) for 1 hr at room temperature before plated onto concanavalin A-coated glass-bottomed Petri dishes (WPI). For each stiffness measurement, an isolated spheroplast was identified and DIC imaged. The image was used to determine the spheroplast diameter. Thereafter, the cantilever wedge was positioned ~20 μm above the spheroplast and lowered at 1 μm·s$^{-1}$ until an upward force of 10 nN was recorded. The cantilever height was maintained for 10 s before the cantilever was raised 10 μm at 1 μm·s$^{-1}$. The cantilever was then lowered onto the same cell, paused and raised successively for seven further cycles. The cantilever height and the force acting on the cantilever were recorded at 40 Hz. The slope (least-squares fit, nN·μm$^{-1}$) of the data points between 3 and 7 nN of force of the downward phase of each cycle was determined. The mean of these slopes was used as the measure of spheroplast stiffness.

## Spheroplasting assays

For budding yeast cells, yYB5978 was grown overnight in YPD (1% yeast extract, 2% peptone, 2% dextrose) at 30℃ and diluted to OD$_{600}$ = 0.1 in 25 mL YPD. Two hours later, mating pheromone (α-factor, Sigma T-6901-1mg) was added at a concentration of 8 ng/mL in YPD containing 20 μg/mL casein as a blocking agent. After 4–5 hr, the shmooed cells were washed into either SCD or SC (for non-starved and starved conditions respectively) and incubated at room temperature for 1 hr. Cells were then spun down and resuspended into either SCD or SC containing 1 M sorbitol and 200 μg/mL zymolyase 100T (Amsbio). Cells were incubated at 30℃ for 1 hr. Spheroplasting efficiency was examined upon cell lysis in water.

For fission yeast cells, KWY6241 was grown overnight in Edinburgh minimal medium + 2% dextrose (EMMD) to log-phase, then washed into either EMMD or EMM (EMMD with no dextrose) and incubated at room temperature for 1 hr. Cells were then spun down and resuspended into either EMMD or EMM containing 1.2 M sorbitol and 5 mg/mL lysing enzmes from *Trichoderma harzianum* (Sigma L1412) and 0.5 mg/mL zymolyase 100T (Ambsio). Cells were incubated at 30℃ for 1 hr. Spheroplasting efficiency was examined upon cell lysis in water.

For experiments in which the cells were returned to dextrose media, the cells were spun down and resuspended into dextrose-containing medium (SCD + 1 M sorbitol for *S. cerevisiae* or EMMD + 1.2 M sorbitol for *S. pombe*).

## Acknowledgements

We thank Ryan Melnyk and John Coates for the BW25113 strain of *E.coli*, Gertien Smits for the pHluorin construct, Philipp Christen for assistance with the lyophilization equipment, Robert Calderon and Chris Niyogi for access to and help with the Coulter Counter, and Fabrice Caudron for help with the yeast shmooing protocol. We also thank Alan Lowe for aiding in the development of our analysis regime and for discussions, Simon Alberti for communicating results prior to publication,

Mikael Backlund for discussions on the biophysical properties of intracellular diffusion and Leon Chan for review of the manuscript and critical discussions. E.D. was supported by an EMBO long-term fellowship (ALTF 182-2010) and L.H. acknowledges support from the William Bowes Research Fellows Program. This work was supported by NIH/NIGMS (R01GM058065 and R01GM101257 to K. W.) and the Swiss National Fonds (SNF 159731).

## Additional information

### Competing interests

KW: Reviewing editor, *eLife*. The other authors declare that no competing interests exist.

### Funding

| Funder | Grant reference number | Author |
|---|---|---|
| National Institute of General Medical Sciences | R01GM058065 | Karsten Weis |
| National Institute of General Medical Sciences | R01GM101257 | Karsten Weis |
| Schweizerischer Nationalfonds zur Förderung der Wissenschaftlichen Forschung | SNF 159731 | Karsten Weis |
| European Molecular Biology Organization | ALTF 182-2010 | Elisa Dultz |

The funders had no role in study design, data collection and interpretation, or the decision to submit the work for publication.

### Author contributions

RPJ, conceived the study, performed the particle tracking experiments, filamentous actin quantification, ATP measurements, intracellular pH measurements, and cell volume measurements, analyzed the results, wrote the manuscript; JHT, conceived the study, performed the vacuole-to-cell volume quantification, the lyophilization cell mass measurements, the AFM stiffness measurements, and the spheroplasting experiments, wrote the manuscript, analysis and interpretation of data; JH, performed the AFM stiffness measurements, analyzed AFM stiffness measurements; ED, developed code for the analysis of particle tracking and intracellular pH measurements and performed nucleus volume measurements, analysis and interpretation of data; CB, established the intracellular pH calibration curve, acquisition of data; LJH, performed the microtubule counting experiments, acquisition of data; SH, performed analysis on the diffusion experiments and helped with the interpretation of the results; DJM, analyzed AFM stiffness measurements; KW, conceived the study, wrote the manuscript, analysis and interpretation of data

### Author ORCIDs

Liam J Holt, http://orcid.org/0000-0002-4002-0861
Karsten Weis, http://orcid.org/000-0001-7224-925X

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
