## [Decision Letter]

[Editors’ note: this article was originally rejected after discussions between the reviewers, but the authors were invited to resubmit after an appeal against the decision.]

Thank you for choosing to send your work entitled "A Conserved Starvation Response Confines Macromolecular Mobility" for consideration at *eLife*. Your full submission has been evaluated by Vivek Malhotra (Senior editor) and two peer reviewers, one of whom is a member of our Board of Reviewing Editors, and the decision was reached after discussions between the reviewers. Based on our discussions and the individual reviews below, we regret to inform you that your work will not be considered further for publication in *eLife*.

While both reviewers found the work of interest and importance, they were unconvinced that the data supported the conclusions. In addition, one reviewer notes that the accompanying Alberti manuscript appears to contradict the observations. In view of these considerations, we do not feel that the manuscript warrants further consideration unless these concerns can be resolved.

*Reviewer #1:*

In this manuscript, the authors include experiments with the goal of providing support to their claim that an increase in macromolecular crowding is what causes a reduction in cellular dynamics in glucose-depleted cells. However, there are technical concerns with these experiments. The reported evidence is unconvincing, as explained below. In fact, a comparison between the data here and those in the Alberti paper argue against crowding being the primary cause for the observed particle confinement. In addition, the concern about the bacterial work was not addressed, and the generalization to bacteria remains unwarranted.

The data shown in Figure 7B and Figure 7—figure supplement 1 suggest that there is something wrong about their FRET-based crowding sensor. The change in crowding under glucose-depletion condition is proposed to be equivalent to a hyperosmotic shock with 2 M sorbitol. This is a massive osmotic shock. GFP diffusion should be severely impacted under these conditions; yet they show that GFP diffusion is unchanged in glucose-depleted cells. The cytoplasmic volume should also be more reduced with 2 M sorbitol than what they report for glucose depletion. Consistent with this, the accompanying paper from the Alberti group reports a large reduction in both cytoplasmic volume and GFP diffusion in cells exposed to 2 M sorbitol. Thus, the crowding in glucose-depleted cells is unlikely to be equivalent to cells exposed to 2 M sorbitol. Could the FRET-based sensor be affected by the difference in pH between the glucose-plus and glucose-depleted conditions? Fluorophores are often sensitive to pH, which would make the measurements unreliable.

In Figure 5, it is argued that "a reduction in cell volume is sufficient to explain the starvation-induced effect on macromolecular mobility". However, it is unclear that the reduction in cytoplasmic volume under their glucose-depletion conditions is equivalent to the reduction in cytoplasmic volume under the reported hyperosmotic shock conditions (> 0.4M NaCl).

Furthermore, the key issue was to determine the cytoplasmic volume and not the cell volume/size (the cytoplasm can shrink more than the cell because of the cell wall).

Also, similarly to the 2 M sorbitol experiments, how is GFP diffusion under the hyperosmotic conditions of high concentrations of NaCl? GFP diffusion should be significantly reduced (e.g., see van den Bogaart et al., Mol Microbiol 2007) in which case it argues that their hyperosmotic shock treatment does not phenocopy their glucose-depletion condition (where GFP diffusion is unaffected), contrary to their claim.

The authors appropriately take into consideration the increase in vacuole size, but they should probably also consider the decrease in nucleus size.

Figure 6: Is the assay to determine cell mass/cell sensitive enough to draw conclusions about glucose-plus and glucose-deprived cells? What is the difference in cytoplasmic volume (after correction for vacuole and nucleus size) between raffinose-plus and glucose-plus cells? Is it similar to the difference between glucose-plus and glucose-depleted cells? This is important to know. If the difference is much larger, it is unclear that the assay would be able to measure a difference in cellular mass between glucose-plus and glucose-depleted cells since the difference in cellular mass between raffinose-plus and glucose-plus cells was only 23%. There is typically a lot of variability in cell number measurements.

Figure 7D: As mentioned above, cytoplasmic volumes and not cell volumes should be measured for DNP-treated *E. coli* cells. Furthermore, the 30-min DNP treatment is too long to be compared to the data in Parry et al., 2014. Cellular composition can change in 30 min, and Parry et al. report an immediate effect on particle dynamics following DNP treatment. The presented evidence is too tenuous to draw conclusions about what happens in bacteria.

As currently presented, the conclusions between the two co-submitted papers appear contradictory.

*Reviewer #2:*

In this paper, the authors track various structures in yeast cells, calculate their mean squared displacements (MSD) and find that these structures have a reduced mobility when the cells are starved ("- Glucose condition"). The authors show that a decrease in ATP or pH to levels similar to those seen during starvation are alone not sufficient to explain the reduced mobility. However, they notice that as they lower pH they not only lower the measured mobility in the cell, they also reduce the cell volume. Surprisingly, when they lower pH to the point where the mobility in the cell is similar to that seen in a starved cell, they find a similar volume reduction as the volume reduction observed for a starved cell. From this, they conclude that the reduced mobility arises from an increased confinement owing to this reduced cell volume. Overall, I think the paper is a bit weak in its current form. The reasons follow:

1) The authors don't really show that a given volume change always corresponds to a given decrease in mobility. They just show that one mobility and cell volume under starvation conditions corresponds to the same mobility and volume under one other condition (8mM K^+^Sorbate). In particular, the authors don't properly correlate a measured change in mobility to a measured change in volume under different conditions. They attempt to do this through the addition of NaCl, but they never report the actual volume change with increasing salt condition (They just say the volume decreases). In addition, they have data (2mM K^+^Sorbate) that indicate a similar degree of cell volume change without a significant decrease in mobility. This contradicts the general conclusion of the paper (see point 2, below).

A) In addition, the authors attribute a decreased mobility under some conditions (loss of actin or microtubules) to effects other than volume change. This would indicate that there are conditions in which mobility is affected without changing cell volume. Therefore, they need more evidence to show that the cell volume decrease and mobility under starvation conditions doesn't just coincidentally coincide with those of 8mM K^+^Sorbate conditions.

2) Currently, the authors only relate the volume of a cell under starved conditions to that of a cell whose volume is reduced while the pH of the cell is also reduced. They should do a test where they only change cell volume and leave all other variables fixed. They attempt to do this through osmotic pressure with the addition of NaCl, but they never actually measure the amount of cell volume reduction under these conditions.

A) In addition, I don't know enough about yeast biology to know what NaCl does to the cell. If it isn't an inert osmotic shock, it may make sense to do a test using an inert crowder like PEG or dextran.

3) The authors conclude that this is a conserved mechanism because they show that the volume of *E. coli* also decreases under conditions where the internal dynamics of *E. coli* show a reduced mobility. However, they don't show that something more definitive like a similar volume decrease leads to a similar decrease in mobility or relate the volume decrease to mobility through some theoretical prediction with regards to molecular crowding. (Again, they just test this one condition). The claim from this limited investigation that this is a conversed mechanism is too strong in my mind.

Overall, I think their study is interesting and a link between mobility and cell volume could be interesting. However, I find that the authors’ claim that the reduced mobility under starvation conditions arises from a reduction in cell volume too strong. The data are suggestive but not conclusive (I think they are over-selling it). In addition, the way the story is presented and the evidence for their conclusion is slightly contradictory indicating that this most likely not the full story.

---

## [Author Response]

[Editors’ note: the author responses to the peer reviewers follow.]

*While both reviewers found the work of interest and importance, they were unconvinced that the data supported the conclusions. In addition, one reviewer notes that the accompanying Alberti manuscript appears to contradict the observations. In view of these considerations, we do not feel that the manuscript warrants further consideration unless these concerns can be resolved. Reviewer #1: In this revised manuscript, the authors include experiments with the goal of providing support to their claim that an increase in macromolecular crowding is what causes a reduction in cellular dynamics in glucose-depleted cells. However, there are technical concerns with these experiments. The reported evidence is unconvincing, as explained below. In fact, a comparison between the data here and those in the Alberti paper argue against crowding being the primary cause for the observed particle confinement. In addition, the concern about the bacterial work was not addressed, and the generalization to bacteria remains unwarranted.*

We respectfully disagree with this reviewer, and as addressed in more detail below, we are confident that our results and conclusions are valid. Regarding the conflicting data with the Alberti paper, we show in our manuscript that the physiological pH drop that occurs upon glucose starvation is neither necessary nor sufficient to recapitulate the confinement of macromolecules that we have observed during this physiological stress. Furthermore, we include evidence (see Figure 7) that the response of yeast cells to the treatment described by Munder et al. (2-deoxyglucose and antimycin A incubation) is different from the response we see in our acute glucose starvation treatment (rapid removal of glucose). We therefore believe that the phenomena being described in these two papers cannot be directly compared. Nevertheless, both papers have in common that they show that yeast cells have the ability to dramatically alter their biophysical properties.

Author response image 1.Comparison of acute glucose starvation and 2-deoxyglucose/antimycin A treatments.(**A**) MSD curves for GFA1 mRNP particles in non-starved cells (blue), cells acutely starved of glucose (red), and cells acutely starved of glucose additionally treated with 20 mM 2-deoxyglucose (2DG) and 10 µM antimycin A (AntA) (green). (**B**) Fluorescence intensity histograms for GFA1 mRNP particles in non-starved (blue), acutely starved (red), and 2DG/AntA-treated cells (green).**DOI:**
http://dx.doi.org/10.7554/eLife.09376.028

We agree that we have only done very limited experiments with *E. coli* and thus our original statement that the nature of the particle confinement in bacteria and yeast is conserved, is not strongly supported in our results. Therefore, while we find our results intriguing, we have removed our argument for the conserved aspect of this phenomenon.

*The data shown in Figure 7B and Figure 7—figure supplement 1 suggest that there is something wrong about their FRET-based crowding sensor. The change in crowding under glucose-depletion condition is proposed to be equivalent to a hyperosmotic shock with 2 M sorbitol. This is a massive osmotic shock. GFP diffusion should be severely impacted under these conditions; yet they show that GFP diffusion is unchanged in glucose-depleted cells.*

We are not sure what evidence the reviewer uses to “believe” that “GFP diffusion should be severely impacted under these conditions” and “that something must be wrong” with the sensor that we have used. The paper that is cited below (van den Bogaart et al., Mol Microbiol 2007) is from work in *E. coli*, and as this reviewer correctly points out it is not possible to compare directly results in yeast with the ones in *E. coli*. Yet, since both reviewers questioned the validity of the sensor, and it has not yet been widely used and confirmed in the literature, we have removed the FRET biosensor results.

*The cytoplasmic volume should also be more reduced with 2 M sorbitol than what they report for glucose depletion.*

We don’t know where the evidence for this argument comes from but we have used 0.4M NaCl and not 2 M sorbitol to successfully mimic the macromolecular confinement observed upon glucose withdrawal.

*Consistent with this, the accompanying paper from the Alberti group reports a large reduction in both cytoplasmic volume and GFP diffusion in cells exposed to 2 M sorbitol. Thus, the crowding in glucose-depleted cells is unlikely to be equivalent to cells exposed to 2 M sorbitol. Could the FRET-based sensor be affected by the difference in pH between the glucose-plus and glucose-depleted conditions? Fluorophores are often sensitive to pH, which would make the measurements unreliable.* We have not seen these results from the Alberti lab however, as pointed out above, we have not used 2 M sorbitol to inhibit the macromolecular diffusion but instead use 0.4 M NaCl. Sorbitol was used as a control for our biosensor experiments. However, as there were concerns by both reviewers we have removed the FRET biosensor results from this submission.

*In Figure 5, it is argued that "a reduction in cell volume is sufficient to explain the starvation-induced effect on macromolecular mobility". However, it is unclear that the reduction in cytoplasmic volume under their glucose-depletion conditions is equivalent to the reduction in cytoplasmic volume under the reported hyperosmotic shock conditions (> 0.4M NaCl).* Measuring the cellular volume decrease under the hyperosmotic shock conditions would indeed be informative. Unfortunately, however, we are technically unable to perform this experiment using our methods because the buffer requirements of the Coulter counter (which needs isotonic concentrations of salt for the electric current measurements) precludes measurements under hyperosmotic shock conditions.

*Furthermore, the key issue was to determine the cytoplasmic volume and not the cell volume/size (the cytoplasm can shrink more than the cell because of the cell wall).* Because the yeast cell wall is permeable to ions (but the plasma membrane is not), the Coulter counter experiments directly measure the cytoplasmic volume, not the cell wall.

*Also, similarly to the 2 M sorbitol experiments, how is GFP diffusion under the hyperosmotic conditions of high concentrations of NaCl? GFP diffusion should be significantly reduced (e.g., see van den Bogaart et al., Mol Microbiol 2007) in which case it argues that their hyperosmotic shock treatment does not phenocopy their glucose-depletion condition (where GFP diffusion is unaffected), contrary to their claim.*

As pointed out above, the experiments reported in van den Bogaart et al., Mol Microbiol 2007 were done in *E. coli* and are thus difficult to compare. We cannot perform single molecule tracking experiments with GFP but we were able to perform FCS experiments with GFP but because the methodology and time scale of the measurements are so fundamentally different compared to the single particle MSD measurements reported here, we prefer not to include these results.

*The authors appropriately take into consideration the increase in vacuole size, but they should probably also consider the decrease in nucleus size.*

Yes, this is a valid point, and we have changed the manuscript to reflect this concern. Taking into account the change in nuclear volume in addition to the change in vacuolar volume after glucose starvation, we now calculate that the accessible cytoplasmic volume decreases by 33%.

*Figure 6: Is the assay to determine cell mass/cell sensitive enough to draw conclusions about glucose-plus and glucose-deprived cells? What is the difference in cytoplasmic volume (after correction for vacuole and nucleus size) between raffinose-plus and glucose-plus cells? Is it similar to the difference between glucose-plus and glucose-depleted cells? This is important to know. If the difference is much larger, it is unclear that the assay would be able to measure a difference in cellular mass between glucose-plus and glucose-depleted cells since the difference in cellular mass between raffinose-plus and glucose-plus cells was only 23%. There is typically a lot of variability in cell number measurements.*

These experiments were repeated three times and statistical tests demonstrate that the results are highly significant. Furthermore, the mass-per-cell measurements showed little variability and high agreement with each other in the biological triplicates. Finally, cell number measurements were performed using a Neubauer hemocytometer, which is a routinely used tool for accurate cell counting measurements.

*Figure 7D: As mentioned above, cytoplasmic volumes and not cell volumes should be measured for DNP-treated E. coli cells. Furthermore, the 30-min DNP treatment is too long to be compared to the data in Parry et al., 2014. Cellular composition can change in 30 min, and Parry et al. report an immediate effect on particle dynamics following DNP treatment. The presented evidence is too tenuous to draw conclusions about what happens in bacteria.*

As mentioned previously, the Coulter counter measures the flow of ions, which can completely permeate the porous bacterial cell wall. Thus, our results do indeed measure cytoplasmic volumes and not cellular volumes. However, we have now reduced the scope of our discussion about the possible conserved nature of the diffusional restriction phenomena in bacteria and yeast.

*As currently presented, the conclusions between the two co-submitted papers appear contradictory.*

As explained earlier, we believe that the effects seen in the Alberti paper are significantly distinct from those presented here due to the considerably different treatments used.

*Reviewer #2: In this paper, the authors track various structures in yeast cells, calculate their mean squared displacements (MSD) and find that these structures have a reduced mobility when the cells are starved ("- Glucose condition"). The authors show that a decrease in ATP or pH to levels similar to those seen during starvation are alone not sufficient to explain the reduced mobility. However, they notice that as they lower pH they not only lower the measured mobility in the cell, they also reduce the cell volume. Surprisingly, when they lower pH to the point where the mobility in the cell is similar to that seen in a starved cell, they find a similar volume reduction as the volume reduction observed for a starved cell. From this, they conclude that the reduced mobility arises from an increased confinement owing to this reduced cell volume. Overall, I think the paper is a bit weak in its current form.*

Obviously, we would like to disagree with this assessment. The perceived weakness (see also below) relates to our incomplete mechanistic understanding of the observed phenomenon (i.e. what causes the observed change in macromolecular diffusion upon glucose starvation). In our view the strength of the paper is the remarkable finding that cells regulate their biophysical properties in order to control rates of intracellular diffusion. We have further strengthened this aspect and now include additional experiments demonstrating that yeast undergo a dramatic change in their visco-elastic properties upon glucose starvation. This includes new atomic force microscopy measurements and a novel assay that reveals that glucose starved cells are significantly more rigid than their non-starved controls.

The reasons follow:

*1) The authors don't really show that a given volume change always corresponds to a given decrease in mobility. They just show that one mobility and cell volume under starvation conditions corresponds to the same mobility and volume under one other condition (8mM K^+^Sorbate). In particular, the authors don't properly correlate a measured change in mobility to a measured change in volume under different conditions. They attempt to do this through the addition of NaCl, but they never report the actual volume change with increasing salt condition (They just say the volume decreases).*

We agree that this would be informative. Unfortunately, however, we are technically unable to perform this experiment using our methods because the buffer requirements of the Coulter counter (which needs isotonic concentrations of salt for the electric current measurements) precludes measurements under hyperosmotic shock conditions. The Coulter counter allowed us to measure tens of thousands of cells and lower throughput visual assays might not be sensitive enough to cleanly differentiate between individual salt concentrations although a significant shrinkage can be observed upon 0.4 M NaCl addition (consistent with prior publications).

In addition, they have data (2mM K^+^Sorbate) that indicate a similar degree of cell volume change without a significant decrease in mobility. This contradicts the general conclusion of the paper (see point 2, below).

We apologize but unfortunately our original supplemental figure was not properly displayed and appeared compressed, which seemed to have led to a confusion. The new Figure 5—figure supplement 1 hopefully clarifies this point. Furthermore, we have removed our results using the crowding sensor.

*A) In addition, the authors attribute a decreased mobility under some conditions (loss of actin or microtubules) to effects other than volume change. This would indicate that there are conditions in which mobility is affected without changing cell volume. Therefore, they need more evidence to show that the cell volume decrease and mobility under starvation conditions doesn't just coincidentally coincide with those of 8mM K^+^Sorbate conditions.*

We are not quite sure whether we understand this point. Our data suggest that the effects on chromatin mobility are an indirect consequence of the breakdown of the microtubule and actin cytoskeleton. It was previously shown (and confirmed by us) that the cytoskeleton breaks down during osmotic shock.

2) Currently, the authors only relate the volume of a cell under starved conditions to that of a cell whose volume is reduced while the pH of the cell is also reduced. They should do a test where they only change cell volume and leave all other variables fixed. They attempt to do this through osmotic pressure with the addition of NaCl, but they never actually measure the amount of cell volume reduction under these conditions.

As discussed previously, the Coulter counter method unfortunately does not permit us to measure the volume of the yeast cells under the hyperosmotic NaCl treatment because the measurement buffer requires an isotonic salt concentration.

*A) In addition, I don't know enough about yeast biology to know what NaCl does to the cell. If it isn't an inert osmotic shock, it may make sense to do a test using an inert crowder like PEG or dextran.*

That is correct. Indeed, high concentrations of sorbitol (2M) have similar effects on macromolecular diffusion as 0.4 M NaCl (data not shown).

*3) The authors conclude that this is a conserved mechanism because they show that the volume of E. coli also decreases under conditions where the internal dynamics of E. coli show a reduced mobility. However, they don't show that something more definitive like a similar volume decrease leads to a similar decrease in mobility or relate the volume decrease to mobility through some theoretical prediction with regards to molecular crowding. (Again, they just test this one condition). The claim from this limited investigation that this is a conversed mechanism is too strong in my mind.*

As discussed in our response to reviewer 1, we have now reduced the scope of our discussion regarding the conserved nature of the mechanism in *E. coli* and yeast.

*Overall, I think their study is interesting and a link between mobility and cell volume could be interesting. However, I find that the authors’ claim that the reduced mobility under starvation conditions arises from a reduction in cell volume too strong. The data are suggestive but not conclusive (I think they are over-selling it). In addition, the way the story is presented and the evidence for their conclusion is slightly contradictory indicating that this most likely not the full story.*

We thank this reviewer for his positive assessment. We would agree that our study is interesting demonstrating for the first time that eukaryotic cells dramatically alter their biophysical and mechanical properties as part of a physiological stress response.